# Aurora kinase A is essential for meiosis in mouse oocytes

**Cecilia S. Blengini**[1,2☯], **Patricia Ibrahimian**[1☯], **Michaela Vaskovicova**[3], **David Drutovic**[3], **Petr Solc**[3†], **Karen Schindler**[1,2]*

**1** Department of Genetics; Rutgers, The State University of New Jersey, Piscataway, New Jersey, United States of America, **2** Human Genetics Institute of New Jersey; Piscataway, New Jersey, United States of America, **3** Institute of Animal Physiology and Genetics of the Czech Academy of Sciences, Libechov, Czech Republic

☯ These authors contributed equally to this work.
† Deceased.
* ks804@hginj.rutgers.edu

**Data Availability Statement:** All relevant data are within the manuscript and its Supporting Information files, or deposited in Dryad: https://datadryad.org/stash/share/

## Abstract

The Aurora protein kinases are well-established regulators of spindle building and chromosome segregation in mitotic and meiotic cells. In mouse oocytes, there is significant Aurora kinase A (AURKA) compensatory abilities when the other Aurora kinase homologs are deleted. Whether the other homologs, AURKB or AURKC can compensate for loss of AURKA is not known. Using a conditional mouse oocyte knockout model, we demonstrate that this compensation is not reciprocal because female oocyte-specific knockout mice are sterile, and their oocytes fail to complete meiosis I. In determining AURKA-specific functions, we demonstrate that its first meiotic requirement is to activate Polo-like kinase 1 at acentriolar microtubule organizing centers (aMTOCs; meiotic spindle poles). This activation induces fragmentation of the aMTOCs, a step essential for building a bipolar spindle. We also show that AURKA is required for regulating localization of TACC3, another protein required for spindle building. We conclude that AURKA has multiple functions essential to completing MI that are distinct from AURKB and AURKC.

## Author summary

Female gametes, oocytes, are uniquely prone to chromosome segregation errors in meiosis I that are associated with early miscarriages. The Aurora protein kinases are essential to control chromosome segregation in all cell types. During mitosis, Aurora kinase A (AURKA) regulates the building of the spindle, the machinery responsible for pulling chromosomes apart. Here, we use a genetic approach to demonstrate that AURKA is essential for meiosis I in mouse oocytes. AURKA is required at multiple steps in meiosis I, first to trigger fragmentation of protein structures that make up the two ends of the meiotic spindle and later to regulate the proper localization of TACC3 to build a normal bipolar spindle. These findings are the first demonstration of distinct Aurora kinase function that cannot be compensated for by the other two homologs. Therefore, this mouse model

vvqSYn5G4GavA58WI2MSjlm-cFiIKgdFnIaTwXbW_MI.

**Funding:** This work was supported by an NIH grant to KS (R01 GM112801), the Inter-Excellence Program award (LTAUSA17097) to PS, and by the award from National Sustainability Program of the Czech Ministry of Education, Youth and Sports (LO1609). CSB and KS received salary support from the NIH grant. MV, DD and PS received salary support from the Inter-Excellence Program. The funders had no role in study design, data collection and analysis, decision to publish, or preparation of the manuscript.

**Competing interests:** The authors have declared that no competing interests exist. Author Petr Solc was unable to confirm their authorship contributions. On their behalf, the corresponding author has reported their contributions to the best of their knowledge.

is excellent tool for pinpointing specific Aurora kinase functions and identifying AURKA target proteins critical for chromosome segregation in meiosis I.

## Introduction

Haploid gametes, which are required for sexual reproduction, are generated through meiosis; a cell division that undergoes two successive rounds of chromosome segregation without an intervening round of DNA replication. First, homologous chromosomes are separated during meiosis I (MI), followed by sister chromatid separation in meiosis II (MII). Errors in MI give rise to aneuploid gametes that, if fertilized, lead to congenital birth defects or embryo development failure [1–3]. Critical to accurate chromosome segregation is the formation of a bipolar spindle apparatus which captures chromosomes and pulls them apart. Therefore, defects in spindle building could cause chromosome mis-segregation and aneuploidy.

In somatic cells, spindles are built from microtubules that nucleate from centrosomes. Centrosomes are cellular structures that form the ends, or poles, of the spindle and are composed of centrioles surrounded by organized layers of pericentriolar material (PCM). However, in mammalian oocytes this process is strikingly different because centrioles are eliminated during oocyte development [4–6]. The elimination of centrioles in oocytes is conserved across taxa from flies, echinoderms to mammals, and it is critical to ensure the sole inheritance of sperm centriole in the zygote [5, 6]. In mouse oocytes, spindle formation depends on multiple microtubule-organizing centers (MTOCs) that lack centrioles (acentriolar MTOCs; aMTOCs) but retain PCM that nucleate microtubules [7–11]. During spindle formation, aMTOCs undergo a series of highly regulated, morphological changes. First, aMTOCs coalesce and fragment into smaller aMTOCs. Next, these small aMTOCs are sorted so that after an intermediate multipolar ball-like formation, they finally cluster into the two poles of the spindle [11, 12]. Perturbation of any of these steps dramatically affects the spindle structure and the interaction between microtubules and chromosomes, which ultimately can alter chromosome segregation. One result of this perturbation is that oocytes fail to complete meiosis because they activate the spindle assembly checkpoint (SAC) that monitors attachment of microtubules to kinetochores and delays anaphase onset until all kinetochores are attached to microtubules [13].

The Aurora kinases (AURK) are a family of serine/threonine protein kinases involved in chromosome segregation, in mitosis and in meiosis [14–16]. This protein family has three members: AURKA, AURKB and AURKC. Most somatic cells express only AURKA and AURKB, but oocytes express all three isoforms. In somatic cells, AURKA localizes to centrosomes and is involved in centrosome maturation and separation [17–19] and microtubule nucleation [20, 21]. However, in female meiosis, two AURKs are needed to build a normal bipolar spindle: AURKA and AURKC [22]. AURKA localizes to aMTOCs [23–25], and may contribute to spindle formation through mechanisms different than those used in mitosis: regulating aMTOC numbers [23, 24, 26], the distribution of aMTOCs into two poles [12, 24] and maintaining spindle pole structure [27, 28]. Furthermore, a recent finding suggests that AURKA activity is required to assemble a liquid-like spindle domain (LISD) composed of several regulatory factors. The LISD is proposed to allow rapid, and localized, protein concentration changes around microtubules during spindle formation [10]. Depletion or inhibition of AURKA in mouse oocytes produces short, disorganized spindles, characterized by over-clustered aMTOCs and loss of the LISD [10, 23, 25, 28]. Consistent with these spindle abnormalities, these depleted or inhibited oocytes fail to complete meiosis and arrest in metaphase I [23, 24]. AURKC also localizes to aMTOCs and contributes to aMTOC clustering into two spindle

poles. Prevention of AURKC from localizing to aMTOCs in mouse oocytes causes multipolar spindles to form and increased rates of aneuploid egg production [22].

In oocytes, the AURKs exhibit complex genetic interactions and compensatory abilities. For example, AURKB is the catalytic component of the chromosome passenger complex (CPC) in mitosis. But, in oocytes, AURKC outcompetes AURKB and takes over this CPC role [29, 30]. Furthermore, oocytes can complete meiosis in the absence of both AURKB and AURKC because AURKA can function in the CPC; this is specific to oocytes because this compensation does not occur in HeLa cells or in spermatocytes [31, 32]. However, although AURKA can compensate, it is not complete because a subset of oocytes arrest in metaphase I with short spindles. These short spindles arise because AURKC is required to outcompete AURKA from CPC-binding to keep AURKA at aMTOCs and ensure appropriate spindle length [31]. Because AURKA and AURKC compete for CPC binding and because a second population of AURKC exists at aMTOCs, we asked if the compensatory abilities of AURKA and AURKC were reciprocal.

To test if AURKC can compensate for loss of AURKA, and to further understand the role of AURKA during meiosis in mouse oocytes, we generated a mouse strain that lacks *Aurka* [33] specifically in oocytes using *Gdf9*-mediated Cre excision [34]. Consistent with AURKA being the most abundant AURK in oocytes [31], we demonstrate that AURKA is essential for oocyte maturation through fragmenting aMTOCs, regulating localization of TACC3 at the spindle and may have an unknown function to promote anaphase onset. Moreover, we demonstrate that AURKB and AURKC cannot compensate for loss of AURKA. Therefore, AURKA is the only Aurora kinase essential for MI in mouse oocytes.

## Results

### Generation and confirmation of mice lacking *Aurka* in oocytes

Because AURKA can function in the CPC in the absence of AURKB and AURKC [31], we asked if similar compensatory functions exist in the absence of AURKA. Prior AURKA studies used small-molecule inhibitors such as MLN8237 and overexpression to investigate AURKA's role in mouse oocyte meiotic maturation which do not allow for compensation studies [10, 23, 24, 26, 28, 31, 35]. To assess compensation and potential AURKA-specific requirements, we deleted *Aurka* (*Aurka^{fl/fl}*) using *Gdf9*-Cre; this conditional allele of *Aurka* has been described elsewhere [32]. *Gdf9* expression begins around day 3 after birth in prophase I-arrested oocytes; these oocytes already completed early prophase I events such as chromosome synapsis and recombination. *Aurka* is therefore deleted in growing oocytes, several weeks prior to completion of chromosome segregation in meiosis I. To confirm that AURKA was depleted from oocytes, we first assessed total AURKA levels by Western blotting. Compared to the AURKA signal in oocytes from wild-type (WT; *Aurka^{fl/fl}*) littermates, the signal in *Aurka* knockout (KO; *Aurka^{fl/fl} Gdf9-Cre*) oocytes was absent (Figs 1A and S1). We also assessed the presence of AURKA at Metaphase I (Met I) by immunocytochemistry. In WT oocytes, AURKA localized to Met I spindle poles. Compared to WT, *Aurka* KO oocytes lacked AURKA signal (Fig 1B and 1C). Finally, we measured the activity of AURKA by immunostaining oocytes with anti-phosphorylated CDC25B-serine 351 (pCDC25B), an AURKA substrate that localizes to spindle poles [36]. Consistent with the loss of polar AURKA, there was no detectable pCDC25B in *Aurka* KO oocytes (Fig 1D and 1E). These data indicate that *Gdf9*-mediated Cre excision of *Aurka* is sufficient to deplete AURKA in mouse oocytes.

### *Aurka*-oocyte knockout mice are sterile

To determine the consequence of deleting *Aurka* in mouse oocytes, we conducted fertility trials. Age-matched WT and KO females were mated to WT B6D2F1/J males of proven fertility

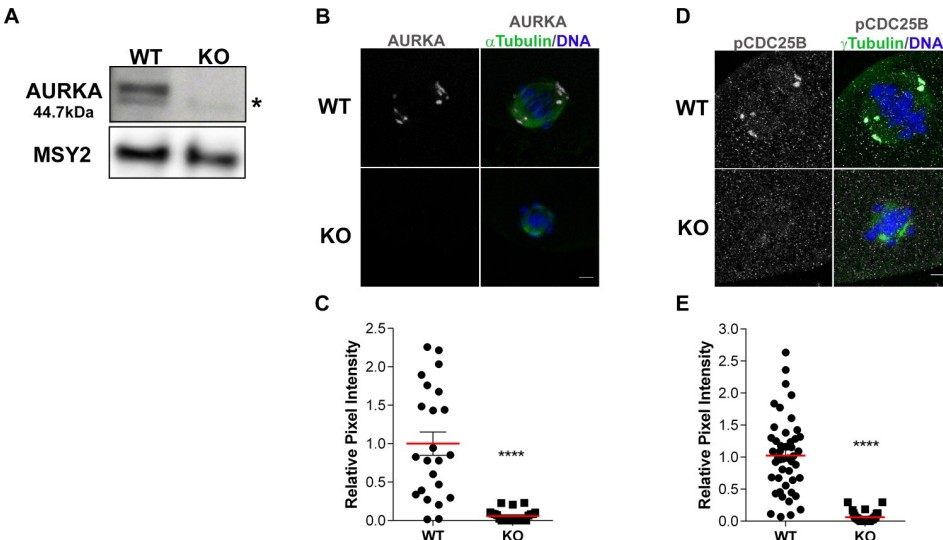

**Fig 1. AURKA is deleted from oocytes. (A)** Western blot detecting AURKA from prophase-I arrested wild-type (WT) and *Aurka* knockout (KO) oocytes (100 oocytes/lane). After stripping the membrane, MSY2 served as loading control. n = 4 animals/genotype/experiment. Asterisk = non-specific band **(B-E)** Localization and activity of AURKA in WT and KO oocytes. **(B)** Representative confocal images of metaphase I oocytes immunostained with antibodies against AURKA (gray), α-Tubulin (green) and DAPI (blue) **(C)** Quantification of AURKA intensity in (B); (Unpaired Student's t-Test, two-tailed, **** $p<0.0001$; number of oocytes, WT: 23; KO: 24) **(D)** Representative confocal images of metaphase I oocytes immunostained with antibodies against phosphorylated CDC25B (gray; pCDC25B), γ-Tubulin (green) and DAPI (blue). **(E)** Quantification of pCDC25B intensity in (D); (Unpaired Students t-Test, two-tailed, **** $p<0.0001$; number of oocytes, WT: 46; KO: 30). Graphs show individual oocyte values plus the mean ± SEM from 2–3 independent experiments. Scale bars: 10μm.

and the numbers of pups born were recorded. We carried out this fertility trial for the time it took WT females to produce 5 litters (~4 months). Compared to WT females that produced ~6 pups/litter, *Aurka* KO females never produced a pup (Table 1). Therefore, in contrast to AURKB and AURKC [30, 31], AURKA is essential for female fertility.

To begin investigations into the cause of sterility, we first evaluated follicle development in histological sections of ovaries from females at different ages (1m, 2m, 6m). Of note, the animals used for histological sampling at 6 months were the females used for the fertility trial because we wanted to ensure that these animals ovulate. Compared to age-matched WT animals, there were no significant differences in the number of follicles at different developmental stages (Fig 2). Importantly, *Aurka* KO ovaries contained corpus luteum (CL), the endocrine-secreting remnant of follicles that release an oocyte, indicating that follicle development and ovulation are normal in these animals. However, *Aurka* KO ovaries from females that were two months old had about one-half of the number of CLs in comparison to WT. Furthermore, although not statistically significant, *Aurka* KO ovaries from older females (six months) also had reduced number of CL (Fig 2C–2F) suggesting that not all oocytes in fully developed

**Table 1. Number of pups, oocytes and cells ovulated from WT and *Aurka* KO females.**

| | WT | | KO | | |
| --- | --- | --- | --- | --- | --- |
| | Mean ± SEM | n | Mean ± SEM | n | p Value |
| Avg. # of pups per litter | 6.25 ± 0.86 | 3 | 0 | 3 | 0.0004 |
| Avg. # of prophase I arrested oocytes | 34.38 ± 5.66 | 13 | 33.92 ± 6.95 | 13 | 0.9593 |
| Avg. # of ovulated cells | 23.75 ± 7.69 | 4 | 18.67 ± 3.84 | 3 | 0.6211 |

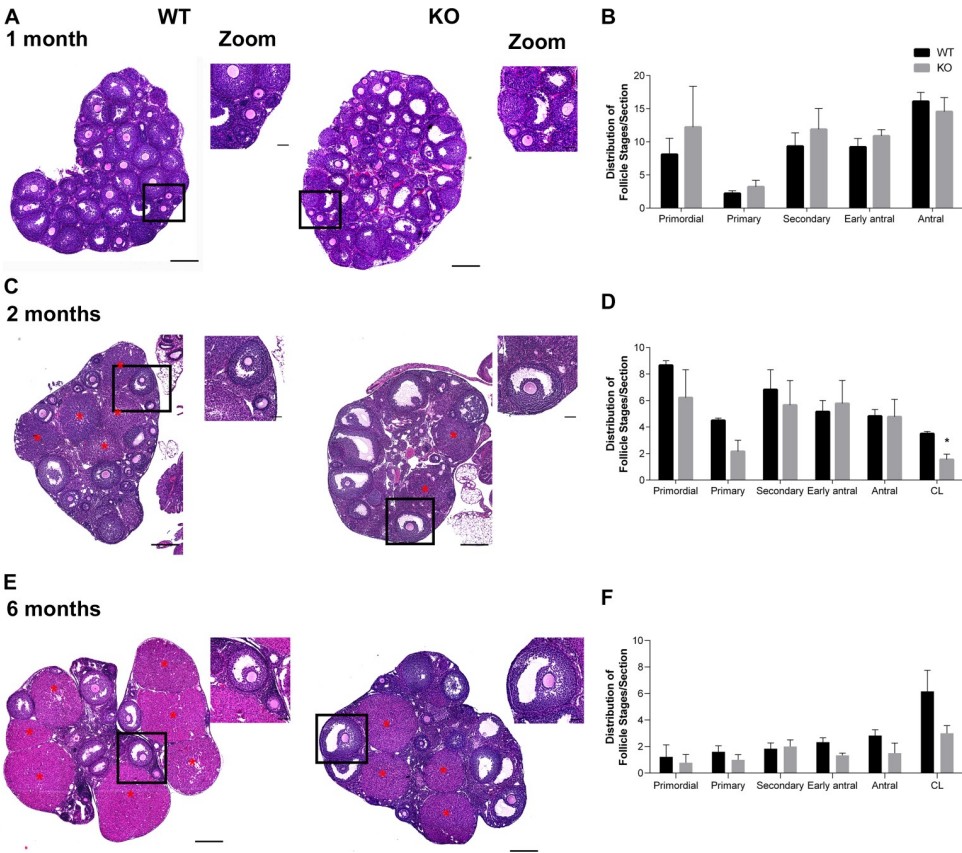

**Fig 2. *Aurka* KO females have normal follicle development. (A, C, E)** Representative images of hematoxylin/eosin-stained ovarian sections from WT and *Aurka* knockout (KO) females from different ages: 1 month (**A**); 2 months (**C**); 6 months (**E**), red asterisks: corpus luteum (CL). The zoom panels highlight commonly observed follicles at each age. (**B, D, F**) Quantification of follicle types from the ovaries represented in (A, C, E) respectively. Follicle numbers were quantified for each ovary and reported as the average number of each type of follicles per section. * p < 0.05. Graph shows the mean ± SEM (1 and 6 months: 3 females/genotype, 2 months: 2 WT; 3 KO). Scale bars: 50μm (zoom panels) and 200μm.

follicles were ovulated. Taken together, these data indicate that the remaining Aurora kinases, AURKB and AURKC cannot compensate for loss of AURKA.

## AURKA has unique functions during meiosis I

We next assessed the quality of the ovulated cells. To harvest cells from oviducts, we had to induce ovulation through hormonal stimulation, which was not used in the fertility trials or histology. To our knowledge, hormone stimulation does not negatively impact meiotic maturation [37]. In this strain background, ~80% of cells in oviducts from WT mice contained polar bodies (Fig 3A and 3B), indicating a completion of meiosis I (MI) and arrest at Metaphase of meiosis II (Met II). In contrast, none of the cells from KO oviducts had polar bodies, and they all were arrested in Met I, indicating a failure to complete MI (Fig 3A and 3B). We note that similar number of cells were obtained from WT and KO oviducts (Table 1).

To identify where in MI *Aurka* KO oocytes were failing, we examined oocytes that were matured *in vitro* for a time in which WT oocytes would reach the Met II arrest. We isolated similar numbers of prophase I-arrested oocytes from WT and KO females (Table 1), consistent with the ovarian reserve not being affected (Fig 2). After maturation, WT oocytes extruded

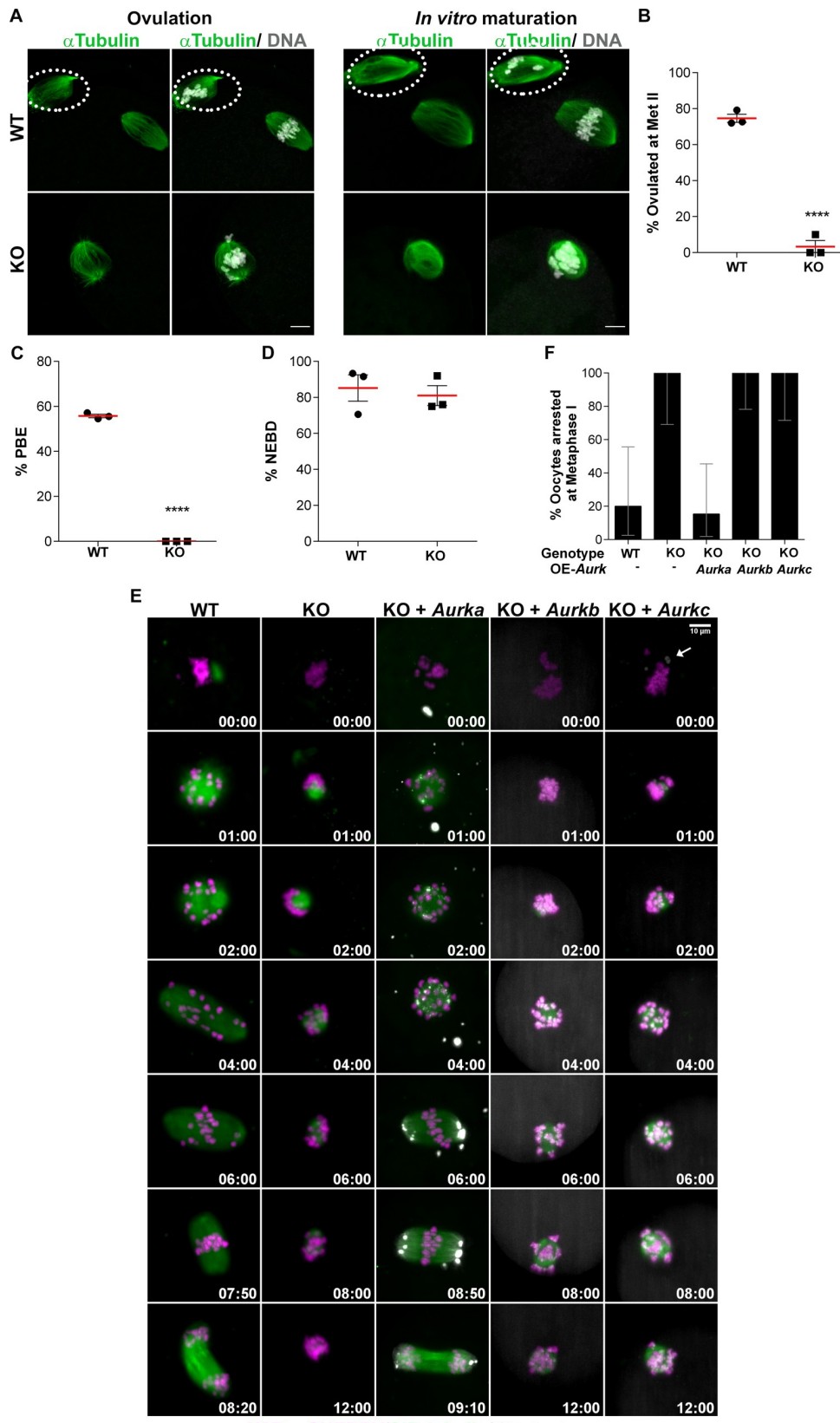

**A** Ovulation *In vitro* maturation

αTubulin αTubulin/ DNA αTubulin αTubulin/ DNA

WT

KO

**B**

% Ovulated at Met II

****

WT KO

**C**

% PBE

****

WT KO

**D**

% NEBD

WT KO

**F**

% Oocytes arrested at Metaphase I

| Genotype | WT | KO | KO | KO | KO |
|---|---|---|---|---|---|
| OE-*Aurk* | - | - | *Aurka* | *Aurkb* | *Aurkc* |

**E**

WT KO KO + *Aurka* KO + *Aurkb* KO + *Aurkc*

10 µm

00:00 00:00 00:00 00:00 00:00

01:00 01:00 01:00 01:00 01:00

02:00 02:00 02:00 02:00 02:00

04:00 04:00 04:00 04:00 04:00

06:00 06:00 06:00 06:00 06:00

07:50 08:00 08:50 08:00 08:00

08:20 12:00 09:10 12:00 12:00

**H2B-mCHERRY**/**SiR-tubulin**/*OE-Aurk*

**Fig 3. AURKA is specifically required in oocytes to complete meiosis I. (A)** Representative confocal images of oocytes and eggs retrieved from oviducts of WT and *Aurka* KO females or oocytes matured *in vitro*. Cells were immunostained with antibodies against α-Tubulin (green) and DAPI (gray) **(B)** Quantification of percentage (%) of cells ovulated at Metaphase II (Met II); (Unpaired Students t-Test, two-tailed, **** p<0.0001). **(C)** Quantification of the % of oocytes that undergo polar body extrusion (PBE) *in vitro* (Unpaired Students t-Test, two-tailed, **** p<0.0001). **(D)** Quantification of the % of oocytes that undergo nuclear envelope breakdown (NEBD) *in vitro* (Unpaired Students t-Test, two-tailed, p = 0.6707). Graphs show the mean ± SEM from 3 independent experiments (3 females/genotype). Scale bars: 10μm. **(E)** Live cell light-sheet imaging of WT and KO oocytes expressing histone H2B-mCHERRY (magenta) and stained with SiR-tubulin (green). Some KO oocytes also expressed exogenous AURKA-EGFP, AURKB-EGFP or AURKC-EYFP (gray). Maximum intensity z-projections and selected time points are shown. White arrow indicates AURKC-EYFP at aMTOC. Scale bars: 10 μm. **(F)** Data in (E) was used to quantify % of oocytes arrested at Metaphase I. Graph shows the mean ± 95% confidence interval. n (WT, KO, KO + *Aurka*, KO + *Aurkb*, KO + *Aurkc*) = 10, 10, 13, 15, 11, respectively.

polar bodies *in vitro*. In contrast, none of *Aurka* KO oocytes extruded polar bodies (Fig 3A–3C). We did not observe a difference between WT and KO oocytes in the percentage of oocytes that resumed meiosis and broke down their nuclear envelopes (Fig 3D). These results indicate that AURKA is essential for meiotic maturation.

To further confirm that AURKB/C cannot compensate for loss of AURKA, we microinjected cRNAs encoding E*gfp* fusions of *Aurka* and *Aurkb*, or E*yfp* fusion of *Aurkc* into *Aurka* KO oocytes. We then visualized spindle formation and chromosome segregation via live cell light-sheet microscopy (Fig 3E and 3F and S1 Movie). As expected, 80% of oocytes from WT mice completed MI, extruded a polar body and reached Met II. In contrast, none of the *Aurka* KO oocytes extruded a polar body and they all remained arrested at Met I. Exogenously expressed AURKA-EGFP localized to MTOCs and decorated MI spindle poles in *Aurka* KO oocytes. Importantly, AURKA-EGFP expression rescued nearly all *Aurka* KO oocytes because they extruded polar bodies and reached Met II (Fig 3F). Ectopic expression of AURKB-EGFP or AURKC-EYFP, however were unable to rescue MI failure and none of these oocytes extruded a polar body. We were surprised that exogenous expression of AURKC could not rescue the ability to complete MI, because a sub-population of AURKC localizes to meiotic spindle poles [22] (Figs 3E and S2) and the AURKs have some overlapping substrate specificity [38]. This failure to rescue suggests that AURKA and AURKC have unique functions that are likely spatially distinct at the poles.

### *Aurka* KO oocytes are defective in MI spindle building

To determine what unique functions AURKA is required for, we next evaluated spindle formation using immunofluorescence staining of fixed oocytes. Inhibition of AURKA with MLN8237 causes MI spindle defects, ranging from bipolar spindles of reduced length and area, spindles with multiple poles and to monopolar spindles [10, 28] (S3A–S3D Fig). We matured oocytes for the time it took the WT oocytes to reach early pro-Metaphase I (pro-Met I) (3h), late pro-Met I (5h), and Met I (7h) stages *in vitro* prior to fixation (Fig 4A). We observed differences between oocytes in early pro-Met I. At this first time point, chromosomes in WT oocytes resolved from one another, consistent with the presence of a microtubule ball that makes transient interactions with chromosomes (Fig 4A and 4B). The microtubule ball was associated with multiple small, γ-Tubulin-positive aMTOCs indicating that aMTOC fragmentation occurred [12]. In contrast, the chromosomes in the majority of *Aurka* KO oocytes did not resolve from one another at early pro-Met I although they did later, suggesting that chromosome resolution in *Aurka* KO oocytes was delayed (Fig 4B). The chromosome morphology appeared normal, and we did not observe a loss of cohesion between homologs or sister chromatids. We did, however, observe fewer and larger γ-Tubulin foci indicating a failure to fragment aMTOCs. Next, when WT oocytes transitioned from pro-Met I to Met I, the

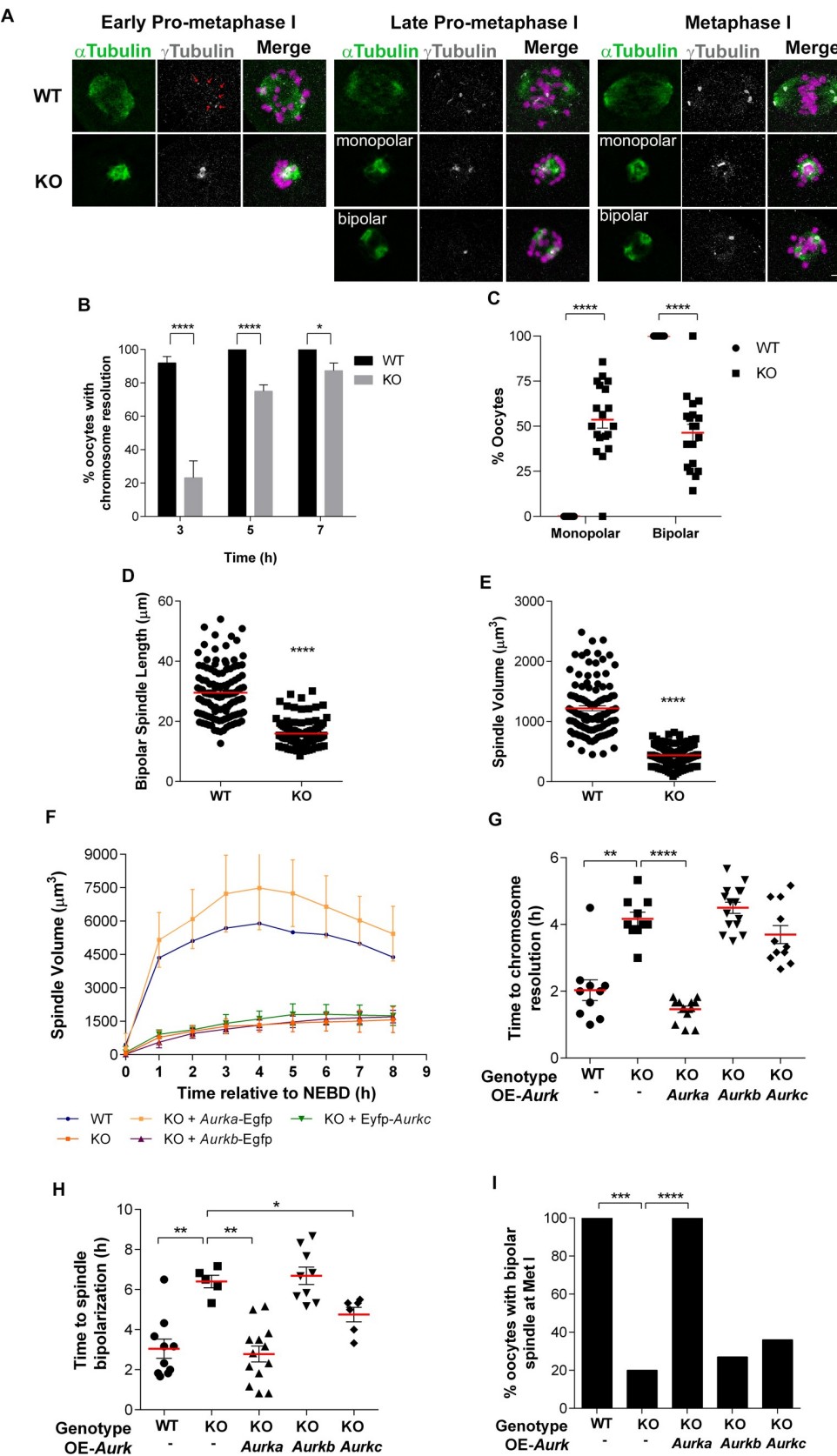

**Fig 4.** *Aurka* **KO oocytes have defects in spindle building. (A)** Representative confocal images of oocytes from WT and *Aurka* KO females matured to different stages of meiosis, as indicated, and immunostained with antibodies against γ-Tubulin (gray), α-Tubulin (green) and DAPI (magenta) Red arrows: aMTOC fragments. **(B)** Quantification of the percentage (%) of oocytes with resolved chromosomes at different meiotic stages (Unpaired Students t-Test, two-tailed, **** p<0.0001; * 0.0196, 6 experimental replicates). **(C)** Quantification of the % of oocytes with a monopolar or bipolar spindle at Metaphase I (Two-way ANOVA; ****p<0.0001, 19 females/genotype). **(D)** Quantification of spindle lengths of bipolar spindles (Unpaired Students t-Test, two-tailed, **** p<0.0001; number of oocytes, WT: 119; KO: 104). **(E)** Quantification of spindle volume (Unpaired Students t-Test, two-tailed, **** p<0.0001; number of oocytes, WT: 113; KO: 143). **(F-I)** Image data from Fig 3E was used for quantification. n (WT, KO, KO + *Aurka*, KO + *Aurkb*, KO + *Aurkc*) = 10, 10, 13, 15, 11, respectively. **(F)** Spindle volume during meiotic maturation. Time is relative to NEBD. **(G)** Time to chromosome individualization and **(H)** spindle bipolarization (Mann-Whitney test, * p<0.05, ** p<0.01, *** p<0.001, **** p<0.0001). **(I)** % of oocytes that had bipolar spindle in Met I (Fisher Exact test, *** p<0.001, **** p<0.0001).

spindles elongated while chromosomes aligned at the Met I plate. Multiple aMTOCs fused together to form two well-defined poles. *Aurka* KO oocytes, however, either had a persistent small microtubule ball with unresolved chromosomes or had elongated spindles. Interestingly, both types of spindles always had 1–2 aMTOCs that did not fragment. When we quantified the distribution of these two spindle phenotypes in KO oocytes, ~55% had monopolar spindles, and ~45% had short bipolar spindles after 7h (Met I) of meiotic maturation (Fig 4C). We also quantified these spindle phenotypes using length and volume measurements. *Aurka* KO oocytes had significantly shorter bipolar spindles (29.54 μm vs 15.92 μm, WT and KO, respectively) and reduced spindle volume (1219 μm$^3$ vs 438.8 μm$^3$, WT and KO, respectively) compared to WT oocytes (Fig 4D and 4E).

To determine if these spindle defects reflect unique AURKA functions, we used these same spindle quantification parameters to assess if the ectopic expression of each of the Aurora kinases can rescue specific steps of meiotic spindle building. Expression of AURKA rescued all the defects: MI spindle volume was restored, chromosomes resolved from one another with WT-like kinetics, and a stable, bipolar MI spindle formed (Figs 3E, 4F and 4J and S1 Movie). Expression of AURKB-EGFP failed to rescue all of these parameters. Interestingly, expression of AURKC-EYFP partially rescued the time in which some *Aurka* KO oocytes formed a bipolar spindle (Fig 4H), although the total number of oocytes that could maintain a bipolar spindle through Met I did not significantly improve (Fig 4I). Taken together, these results suggest that AURKA is uniquely needed for aMTOC fragmentation and building a bipolar MI spindle.

## AURKA is required to fragment aMTOCs through PLK1 and for TACC3 localization

Because our analyses showed that *Aurka* KO oocytes are defective in aMTOC fragmentation, we further investigated this phenotype using high-resolution light-sheet microscopy (S2 Movie). Oocytes expressed H2B-mCherry, CDK5RAP2-Egfp and were stained with a fluorogenic drug, SiR-tubulin, for visualization of chromosomes, aMTOCs, and microtubules, respectively (Fig 5A). Because of the time it took to remove the prophase I-arresting drug from the culture medium and starting the imaging program, we started live imaging 40–50 minutes after meiotic maturation was induced. This timing corresponded to 10–20 minutes prior to nuclear envelope break down (NEBD) in WT oocytes. At this time in control oocytes, one large dominant aMTOC and multiple small aMTOCs were present in the cytoplasm and a few smaller aMTOCs were observed in the perinuclear region (Fig 5A and 5B). When oocytes exited prophase I, as marked by NEBD and chromosome condensation, the majority of cytoplasmic aMTOCs including the dominant aMTOC, moved toward the condensing chromosomes and fragmented. As a result of fragmentation, we observed a reduction in the median size of the major aMTOCs and a subsequent increase in aMTOC numbers (Fig 5A and 5B). As

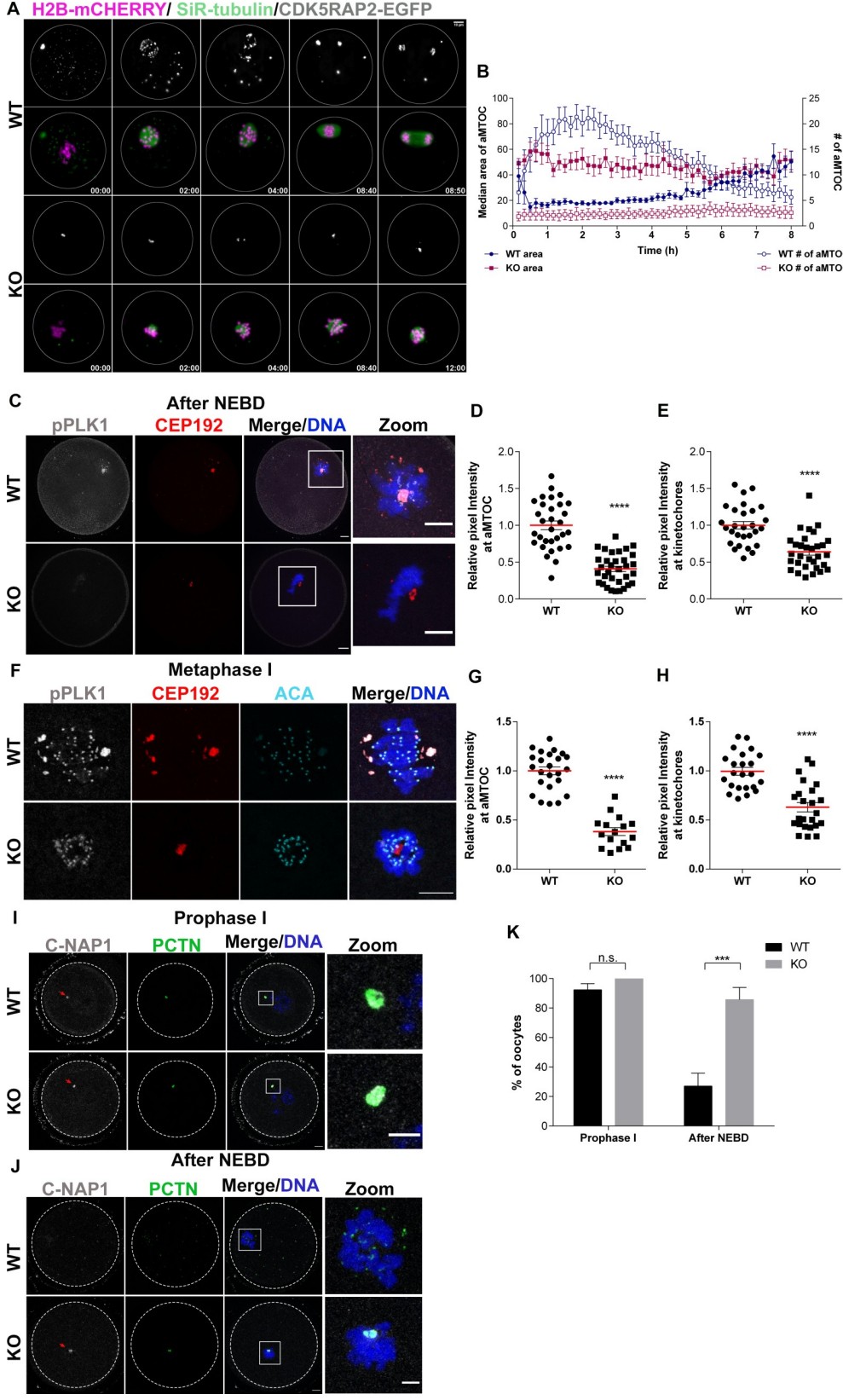

**Fig 5. *Aurka* KO oocytes fail to fragment MTOCs. (A)** Representative images of maximum intensity z-projections from WT and *Aurka* KO oocytes matured live using light-sheet microscopy. Oocytes expressed CDK5RAP2-EGFP (MTOCs, gray) and H2B-mCherry (DNA, magenta) while incubated with SiR-tubulin (spindle, green) are shown. Time points are relative to time after nuclear envelope breakdown (h:min). **(B)** Quantification of median area of aMTOCs (closed symbols) and the number of aMTOC (open symbols) over time from WT (blue) and *Aurka* KO (red) oocytes in (A). Time points are relative to the start of the live imaging. **(C, F)** Representative confocal images of oocytes from WT and KO females after NEBD (C) or at Metaphase I (F) immunostained with antibodies against phosphorylated PLK1 (pPLK1, gray), CEP192 (red) and DAPI (blue); panels in F also include anti-centromeric antigen (ACA; cyan) to mark kinetochores. **(D)** Quantification of pPLK1 intensity at aMTOCs at NEBD (Unpaired Students t-Test, two-tailed, **** p<0.0001; number of oocytes, WT: 31; KO: 33). **(E)** Quantification of pPLK1 intensity at kinetochores at NEBD (Unpaired Students t-Test, two-tailed, **** p<0.0001; number of oocytes, WT: 28; KO: 31). **(G)** Quantification of pPLK1 intensity at aMTOCs at Met I (Unpaired Students t-Test, two-tailed, **** p<0.0001; number of oocytes, WT: 24; KO: 16). **(H)** Quantification of pPLK1 intensity at kinetochores at Met I (Unpaired Students t-Test, two-tailed, **** p<0.0001; number of oocytes, WT: 23; KO: 24). **(I-J)** Representative confocal images of oocytes from WT and KO females at prophase I (I) and after NEBD (J) immunostained with antibodies against C-NAP1 (gray), PCTN (green) and DAPI (blue). **(K)** Quantification of % of oocytes with C-NAP1 localized at aMTOCs (Unpaired Students t-Test, two-tailed, **** p<0.0001; number of oocytes, Prophase I, WT: 35; KO: 24; after NEBD, WT: 20 KO: 25).

WT oocytes transited from pro-Met I to Met I, aMTOCs sorted, spindles elongated and finally MTOCs coalesced to form two spindle poles. During this time, there is a measurable increase in aMTOC size and a reduction in aMTOC foci numbers (Fig 5A and 5B). In contrast, and consistent with our previous result (Fig 4A), *Aurka* KO oocytes lacked multiple cytoplasmic aMTOCs and had only 1–2 large aMTOCs at the time of NEBD (Fig 5B). *Aurka* KO oocytes never fragmented the large aMTOCs (0/16 KO vs 12/12 WT) and therefore both the size and numbers remained constant (Fig 5A and 5B, and S2 Movie). Moreover, after WT oocytes progressed past Met I, we observed that ~73% of the *Aurka* KO oocytes maintained their spindle configuration. Therefore, the monopolar spindle defects observed at Met I were not due to a delay in spindle building but rather was a terminal phenotype.

To understand the role of AURKA in aMTOC fragmentation, we evaluated the aMTOC regulatory pathway in more detail. First, we compared the number of aMTOCs in prophase I-arrested oocytes after fixation and immunostaining. Both in WT and KO groups we found 1–2 large aMTOCs (S4 Fig), suggesting that aMTOC defects do not occur during oocyte growth but, instead the first defect in *Aurka* KO oocytes is the inability to fragment MTOCs upon exiting from prophase I.

Similar to *Aurka* KO oocytes, *Plk1* KO oocytes also arrest in MI with short spindles and have deficiencies in fragmenting aMTOCs [39]. Because AURKA can activate PLK1 via phosphorylation of Threonine 210 [40], we reasoned that AURKA functions upstream of PLK1 in mouse oocytes. To test this hypothesis, we performed immunocytochemistry to detect the activated form of PLK1 (pPLK1) in WT and *Aurka* KO oocytes after NEBD and at Met I. Consistent with our hypothesis, PLK1-T210 phosphorylation was significantly decreased in *Aurka* KO oocytes at aMTOCs by ~60% (Fig 5C, 5D, 5F and 5G) at both stages; however, levels of total PLK1 at aMTOCs remained unchanged at NEBD and increased at Met I (S5A–S5C Fig). But, surprisingly, pPLK1 levels were also significantly reduced by ~40% at kinetochores in *Aurka* KO oocytes (Fig 5C, 5E, 5F and 5H) despite an apparent increase in total localized PLK1 by Met I (S5B–S5D Fig). The reduction of chromosomal pPLK1 likely reflects an undetectable population of AURKA at kinetochores, a population that has been documented in mitotic cells [41].

An essential step for aMTOC fragmentation is the dissociation of the centrosomal cross linker protein C-NAP1 from aMTOCs, a step which requires PLK1 activity [12]. We next evaluated C-NAP1 localization to confirm PLK1-dependent fragmentation defects. At prophase I, both WT and *Aurka* KO oocytes contained aMTOC-localized C-NAP1 (Fig 5I and 5K).

Consistent with reduced PLK1 activity, C-NAP1 persisted at aMTOCs after NEBD in *Aurka* KO oocytes whereas it disappeared in WT oocytes (Fig 5J and 5K). These data suggest that AURKA regulates MTOC fragmentation by phosphorylating and thereby activating PLK1 after NEBD.

Next, we used super resolution microscopy to understand the consequences of the failure of aMTOC fragmentation on Met I spindle pole structure by assessing PCM components pericentrin (PCNT) and CEP215 [28]. WT oocytes had two poles, each of which had the characteristic broad MTOC structure of Met I oocytes. In contrast, some *Aurka* KO oocytes lacked a visible aMTOC (27%) or others had one hyper-condensed spindle pole, with reduced width and volume (Fig 6A–6C). These results are consistent with previous findings that show a collapse of spindle poles after AURKA inhibition [28]. Because we observed changes in the structure of spindle poles in *Aurka* KO oocytes, we used STED-based microscopy to evaluate if AURKA is required for the organization of PCM components. We evaluated the levels of colocalization between CEP215 and PCNT by measuring the covariance in the signal intensity between the two proteins (Pearson coefficient) and by measuring the proportion of overlap of one protein with respect to the other (Manders coefficient). However, we did not observe statistically significant differences in the patterns of colocalization between WT and KO oocytes (Fig 6D–6G), suggesting that the arrangement of these PCM components is not controlled by AURKA.

Next, we assessed microtubule-associated proteins. In mouse oocytes, kinesin-5 (Eg5; KIF11) is required for spindle bipolarization through sliding anti-parallel microtubules apart to facilitate separation of aMTOC fragments [12], and its inhibition causes monopolar spindles, similar to the monopolar phenotype in KO oocytes. We therefore examined KIF11 at Met I aMTOCs. We observed nearly a two-fold enrichment of KIF11 levels at spindles poles in *Aurka* KO oocytes compared to WT (Fig 6H and 6I), an increase which likely reflects the aMTOC size. Therefore, loss of AURKA does not affect KIF11 localization. Because KIF11 is an AURKA substrate it is possible that this enrichment reflects an accumulation of KIF11 in a non, or reduced function state.

Finally, recent evidence suggests that the MI spindle has phase-separated structures that aid in its formation [10]. A key component and marker of this liquid-like spindle domain (LISD) is TACC3, a known AURKA substrate [21, 38, 42]. Consistent with this connection, inhibition of AURKA with MLN8237 disrupted the localization of TACC3 in mouse oocytes. We therefore evaluated the localization of TACC3 in *Aurka* KO oocytes. Upon probing WT and *Aurka* KO oocytes with anti-TACC3 antibodies, we found loss of TACC3 signal (Fig 6J and 6K). Taken together, these results indicate that AURKA is required to build a proper MI spindle through controlling the initial step of fragmenting MTOCs and regulating the localization of TACC3.

## *Aurka* KO oocyte Met I arrest is independent of the SAC

Finally, we evaluated a potential mechanism that would cause the failure to extrude a polar body when many oocytes had small, bipolar spindles. One possibility is the spindle assembly checkpoint (SAC). Insufficient tension between kinetochores and microtubules activates an error-correction pathway involving AURKB/C which triggers detachment of MTs from kinetochores. This loss of kinetochore-microtubule (K-MT) attachments activates the SAC [43] and results in cell-cycle arrest preventing anaphase I. We suspected the arrest in *Aurka* KO oocytes was due to a lack of tension from both monopolar and short spindles. We investigated the strength of the SAC in Met I by evaluating MAD2 signals at kinetochores (Fig 7A). When normalized to kinetochore signal, *Aurka* KO oocytes had significantly higher MAD2 than WT

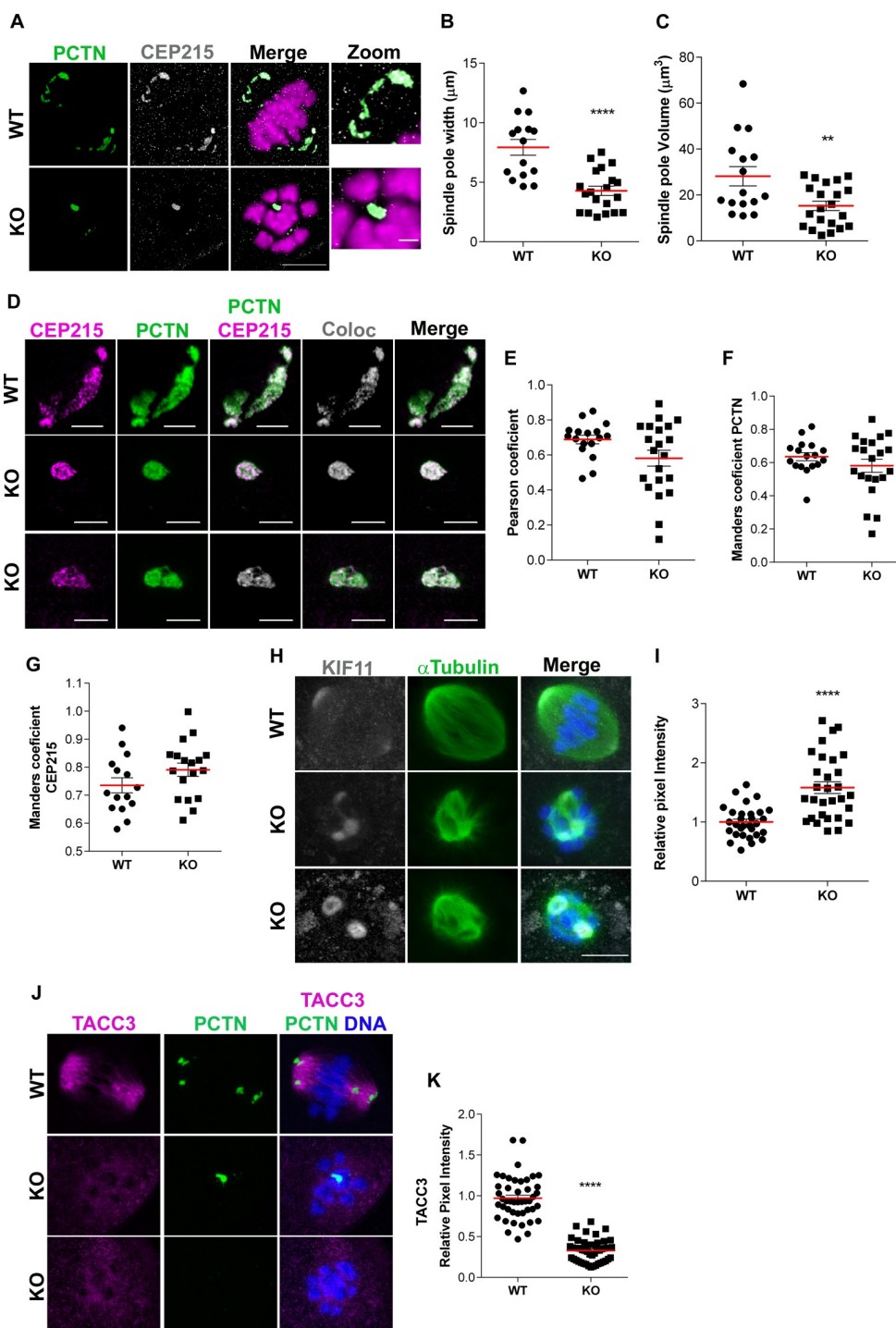

**Fig 6. Analysis of aMTOCs and microtubule associated proteins.** (**A**) Representative images of Metaphase I oocytes from WT and KO females visualized with super-resolution microscopy (Lightning) and immunostained with antibodies against Pericentrin (PCTN, green), CEP215 (gray) and DAPI (magenta). Scale bars: 10μm and 2μm. (**B**) Quantification of spindle pole width in A (Unpaired Students t-Test, two-tailed, **** p<0.0001; number of oocytes, WT: 15; KO: 20). (**C**) Quantification of spindle pole volume in A (Unpaired Students t-Test, two-tailed, ** p = 0.0051; number of oocytes, WT: 16; KO: 21). (**D**) Representative images from STED microscopy of aMTOCs of WT and KO oocytes at Metaphase I immunostained with antibodies against Pericentrin (PCTN, green), CEP215 (magenta); colocalization specific pixels (gray). (**E**) Quantification of Pearson coefficient. (**F, G**) Quantification of Manders coefficient for Pericentrin (Unpaired Students t-Test, two-tailed, p = 0.2736; number of oocytes, WT: 17 A KO: 21) and

CEP215 (Unpaired Students t-Test, two-tailed, p = 0.129; number of oocytes, WT: 15; KO: 18 respectively. Scale bars: 3 μm. **(H)** Representative confocal images of oocytes from WT and KO females at Metaphase I immunostained with antibodies against KIF11 (gray), α-Tubulin (green) and DAPI (blue). Scale bars: 10μm. **(I)** Quantification of KIF11 intensity in H (Unpaired Students t-Test, two-tailed, **** p<0.0001; number of oocytes, WT: 30; KO: 30) **(J)** Representative confocal images of oocytes from WT and KO females at Metaphase I immunostained with antibodies against TACC3 (magenta), PCTN (green) and DAPI (blue). Scale bars: 10μm and 2μm. **(K)** Quantification of TACC3 intensity in J (Unpaired Students t-Test, two-tailed, **** p<0.0001; number of oocytes, WT: 45; KO: 49).

oocytes did (Fig 7B). These data suggest persistent, or elevated, SAC activity in KO oocytes, likely due to a defective spindle and the loss of tension.

Next, to assess whether the Met I arrest in *Aurka* KO oocytes is solely due to persistent SAC activation, we treated oocytes with reversine to inhibit monopolar spindle 1 (MPS1) kinase, a protein required for initiating the SAC signaling complex [44, 45]. We monitored chromosome segregation, spindle formation and polar body extrusion by light-sheet live cell imaging (S3 and S4 Movies). As a read-out of Anaphase-Promoting Complex/Cyclosome (APC/C) activity, we also monitored the destruction of securin-EGFP (Fig 7C). Ninety-five percent of WT oocytes rapidly degraded securin-EGFP (Fig 7C–7E) before anaphase I and extruded the first polar body (Fig 7F). Anaphase I onset occurred ~9h post-NEBD in this imaging system (Fig 7G). In contrast, all *Aurka* KO oocytes remained arrested at Met I (Fig 7F) and had only minor decreases (~10%) in securin-EGFP demonstrating minimal APC/C activity (Fig 7D and 7E). Note that in the one WT oocyte that remained arrested in Met I (WT MI), a similar minor decrease in securin-EGFP also occurred (Fig 7D and 7E). As expected, in WT oocytes, reversine-treatment accelerated the onsets of both securin-EGFP destruction (Fig 7D and 7E) and anaphase I by 2-3h (Fig 7G); all oocytes extruded the first polar body (Fig 7F). Although reversine-treatment restored securin-EGFP destruction in *Aurka* KO oocytes, this restoration was nearly 2h slower than in WT (Fig 7D and 7E). Thirty-six percent of *Aurka* KO oocytes treated with reversine entered Anaphase I, but there was a ~ 4h delay compared to WT oocytes treated with reversine, (Figs S6A and S6B, S4 Movie), and only one-half (18% of the total) extruded the polar body. The remaining one-half either did not extrude the polar body or they had cytokinesis failure and retracted the polar body into the cytoplasm (Fig 7F; PB error). To our surprise, 64% of the oocytes did not enter Anaphase I and did not extrude a polar body (Fig 7F). Importantly, regardless of the polar body extrusion outcome, the APC/C activities in all WT and *Aurka* KO oocytes treated with reversine were similar (Fig 7E). These data suggest that the Met I arrest in the majority (64%) of *Aurka* KO oocytes treated with reversine cannot be explained by insufficient APC/C activity. Therefore, these data suggest that the SAC is not the sole mediator of the Met I arrest in *Aurka* KO oocytes and suggest that AURKA has an additional role in promoting Anaphase I onset.

In summary, we conclude that AURKA is the only Aurora kinase in mouse oocytes that is essential for fertility and MI [30, 31] (Fig 8). Its unique functions include, at least, initiating MTOC fragmentation through activation of PLK1 and spindle formation through regulating TACC3. These functions are essential for spindle building and completion of MI to generate a healthy, euploid egg.

## Discussion

Together with our previous description of *Aurkb* and *Aurkc* double knockout oocytes [31], we demonstrate here that AURKA is the only essential Aurora kinase required for mouse female fertility and oocyte meiotic maturation. Although these KO females are sterile, they do ovulate, albeit MI-arrested oocytes. In *Aurkc*[-/-] oocytes, AURKA and AURKB compensate, and in *Aurkb*[-/-] oocytes, AURKA and AURKC activities are up-regulated. Furthermore, in double

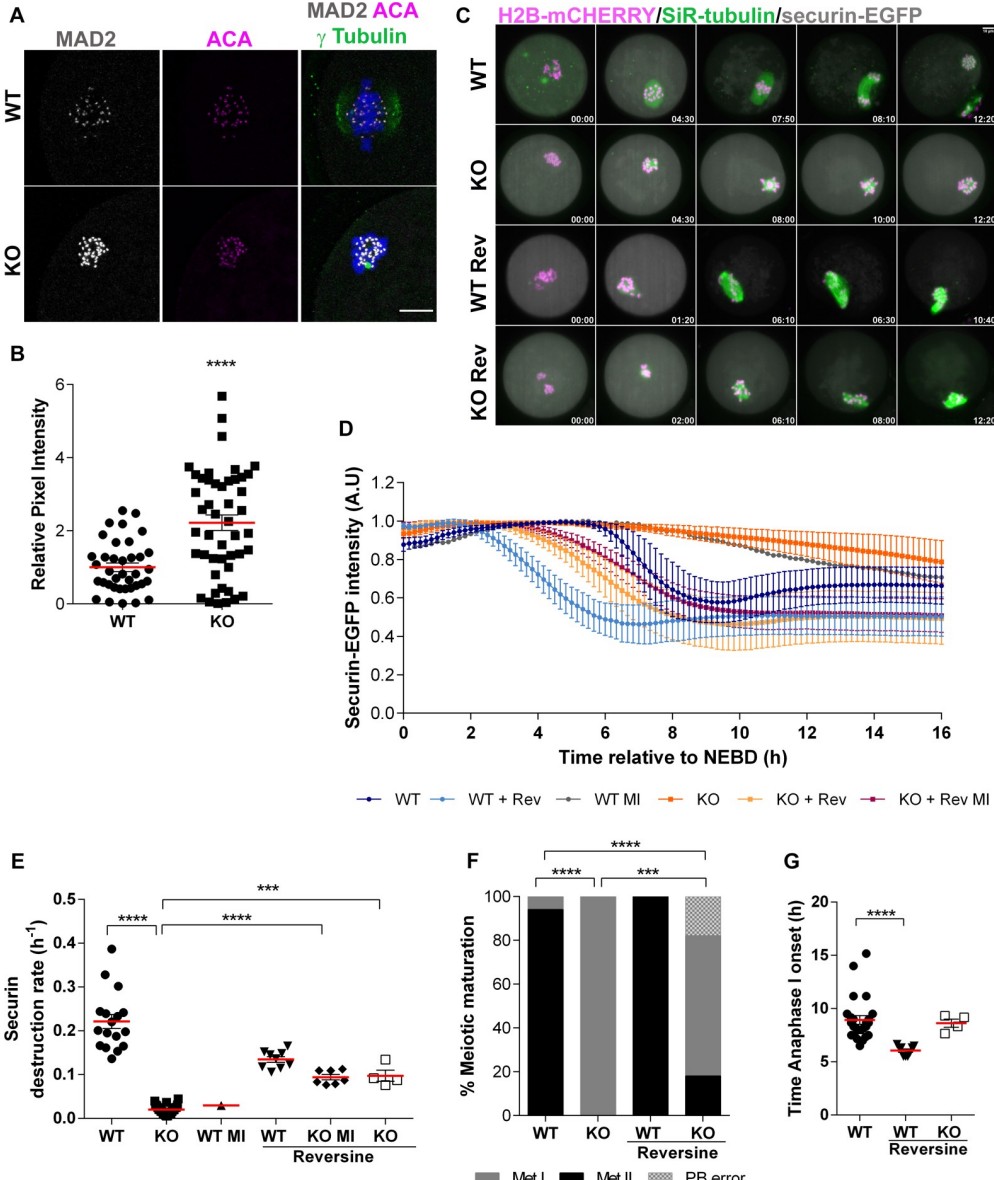

**Fig 7. *Aurka* KO oocyte arrest is SAC independent. (A)** Representative confocal images of oocytes from WT and *Aurka* KO females at Metaphase I immunostained with antibodies to detect centromeres (anti-centromeric antigen (ACA) (magenta)), MAD2 (gray) and chromosomes (DAPI (blue)). **(B)** Quantification of MAD2 intensity at kinetochores in (A) (Unpaired Students t-Test, two-tailed, **** $p < 0.0001$; number of oocytes, WT: 37, A KO: 47). Scale bars: 10μm. **(C)** Live light-sheet imaging of oocytes expressing securin-EGFP (grey), H2B-mCherry (magenta, chromosomes) and stained with SiR-tubulin (green, microtubules) +/- 1μM reversine (Rev) treatment. Maximum intensity z-projection images are shown. Time relative to NEBD. Scale bar = 10 μm. **(D-G)** Data from (C) was used for analysis. Number of oocytes, WT: 18, KO: 24, WT + reversine: 9, KO + reversine: 11. **(D)** Normalized intensities of cytoplasmic securin-EGFP signals. For normalization, maximum securin-EGFP signal in each oocyte was set to 1. Average +/- SD are shown. **(E)** Rate of securin-EGFP destruction ($h^{-1}$) (Mann Whitney Test, ** $p < 0.001$, **** $p < 0.0001$. **(F)** Proportion of WT and KO oocytes +/- 1μM reversine that reached different phases of meiosis (Met I–metaphase I, Met II–metaphase II, PB error–polar body extrusion retraction) after 16 h of time-lapse imaging (Likelihood Test, *** $p < 0.001$, **** $p < 0.0001$). **(G)** Anaphase I onset (hours relative to NEBD), which was defined as the first time point when segregation of chromosomes was detected (Mann Whitney Test, **** $p < 0.0001$).

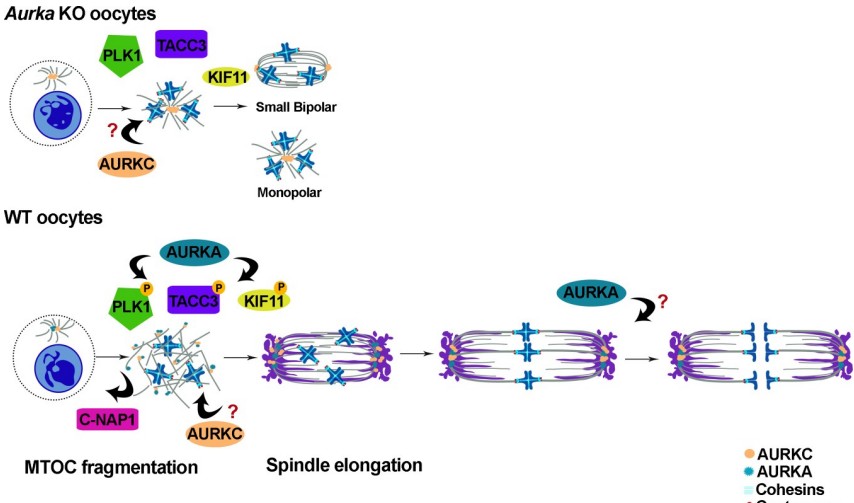

**Fig 8. Schematic comparing WT and *Aurka* KO MI events.** In *Aurka* KO oocytes, AURKC still localizes to aMTOCs but phosphorylated PLK1 is reduced, C-NAP1 persists at aMTOCs and aMTOCs fail to fragment. TACC3 does not localize properly. Some spindles are monopolar, but other spindles can become bipolar, but they are short. The result is an MI arrest. In WT oocytes, AURKA and AURKC localize to aMTOCs, but likely in distinct regions. AURKA is required to phosphorylate PLK1 to induce C-NAP1 release from aMTOCs and initiate aMTOC fragmentation and likely phosphorylates KIF11 and TACC3.

*Aurkb*/*Aurkc* knockout oocytes AURKA compensates [30, 31]. Intriguingly, there is no compensatory mechanism for loss of *Aurka*. Specifically, we show that AURKA is needed at the beginning of meiotic resumption for spindle building. AURKA is required for full PLK1 activation to initiate MTOC fragmentation through inducing C-NAP1 release from aMTOCs and regulates TACC3 localization, likely through phosphorylation, to induce spindle bipolarization. Surprisingly, if the SAC is satisfied, AURKA is also required for Anaphase I onset through an unknown function (Fig 8). Collectively, these data imply AURKA-specific substrates or regulatory partner binding that cannot be carried out by the other 2 Aurora kinases.

Substrate phosphorylation by the AURKs is regulated in at least three ways: 1) activation via autophosphorylation, 2) binding to regulator proteins, and 3) phospho-site consensus motifs. Aurora kinase activity depends on T-loop autophosphorylation and binding to regulatory proteins such as TPX2 and INCENP. These regulatory proteins dictate the subcellular localization of the kinases where they can then access their substrates [46–50]. AURKB and AURKC bind INCENP and function in the CPC at chromosomes and kinetochores, whereas AURKA binds MT-binding proteins like TPX2 and functions on spindles and at spindle poles (MTOCs), where AURKA complexes with PCM proteins exist. The binding affinities for these regulatory proteins are governed by the hydrophilicity of an amino acid in kinase subdomain IV [51, 52]. Substitution of this amino acid in AURKA changes the TPX2-dependent polar localization to INCENP-dependent kinetochore localization. This change in localization allows AURKA to compensate in *AURKB*-depleted HeLa cells. Interestingly, the reciprocal residue alteration in AURKB did not facilitate TPX2-binding, and AURKB therefore cannot carry out AURKA function possibly because it cannot activate upon TPX2 association like AURKA can [53]. In our mouse oocyte studies, we observed similar results: AURKA can carry out CPC functions [31], but AURKB/C cannot carry out AURKA aMTOC functions (Fig 8). Importantly, and different from the HeLa cell experiments, the AURKA-CPC function occurs without amino acid substitution. We speculate that this ability arises because AURKA is the most

abundant of the three AURKs in oocytes and there is therefore likely a soluble pool of free AURKA available to bind INCENP when competition is absent.

The third way Aurora kinase phosphorylation regulation is controlled is through sequence specificity for substrates [54, 55]. SILAC-based phosphoproteomics of HeLa cells, revealed that there are many hundreds of AURKA and AURKB substrates and that their phospho-site consensus motifs are similar but distinct. For example, ~91% of AURKA-dependent phospho-peptides contain a R-R-X-p[S/T] motif, whereas only 8% of the AURKB-dependent phospho-peptides contain this motif. Instead, most AURKB-dependent phospho-peptides contained [R/K]-p[S/T]. Therefore, although the motifs are similar, AURKA prefers an arginine at the -2 position and does not prefer a basic amino acid at the -1 position [38]. However, we demonstrated that when AURKA is the sole Aurora kinase in mouse oocytes it can compensate, indicating that AURKA substrate specificity is flexible. This flexibility is consistent with spindle-pole-localized AURKA triggering MT depolymerization at kinetochores [35, 56], a role that AURKB executes at centromeres, and therefore likely through the same protein substrates. In contrast, because AURKB and AURKC cannot compensate for loss of AURKA, even when overexpressed, this consensus-motif flexibility may not be shared. Alternatively, if AURKA occupies part of the proposed phase-separated spindle domain, it is possible that the regulatory protein that dictates this particular localization cannot bind and/or activate AURKB/C. Additional subcellular targeting of these kinases could help resolve these mechanistic questions.

Because of the number of possible AURKA substrates, it is likely that KO spindle phenotypes arise from a massive change in substrate phosphorylation and downstream function. For example, we show that PLK1 activity is reduced in *Aurka* KO oocytes. PLK1 is a known AURKA substrate. It is not known if AURKC can phosphorylate PKL1, but given that some phosphorylation persisted, it is likely that PLK1 is a shared substrate, a possibility that is worth future investigation. PLK1 is required to promote mitotic entry and centrosome maturation through phosphorylation, one substrate being AURKA in a positive feedback loop [40, 57–59]. *Plk1* knockout in mouse oocytes and oocytes where PLK1 was inhibited share many phenotypes with the *Aurka* KO oocytes [39, 60]. These phenotypes include sterility, MI arrest with short spindles, an inability to fragment aMTOCs, and loss of localized TACC3. However, when PLK1 was inhibited in mouse oocytes, γ-tubulin and pericentrin, were absent from aMTOCs, suggesting that PLK1 is needed to maintain the composition of these structures [60]. Phenotypic differences exist between inhibition of PLK1 and our knockout oocytes [60]. Here, we show that only 27% *Aurka* KOs lack an aMTOC close to the spindle, whereas inhibition of PLK1 abolished MTOCs. This difference can be explained by complete inhibition vs the 60% reduced activation at MTOCs in the *Aurka* KO background. In KO oocytes with aMTOCs, they were smaller and hyper-condensed compared to WT aMTOCs. Although PLK1 and AURKA are part of the same pathway controlling aMTOC fragmentation, there are some PLK1-specific functions; such as, chromosome condensation by regulating localization of condensin and cohesin [39]. *Aurka* KO oocytes can condense the chromosomes, and although they are delayed in resolving individual chromosomes during meiotic maturation, they eventually do.

KIF11 is required for the fragmentation step which occurs after the nuclear envelope breaks down [12], and is a known AURKA substrate in *Xenopus oocytes* [61]. We observed that *Aurka* KO oocytes have increased KIF11 levels, highly enriched at the non-fragmented aMTOCs. It is therefore likely that a failure to phosphorylate KIF11 can explain the subset of oocytes that retain a monopolar spindle, although AURKA phosphorylation of KIF11 has not formally been demonstrated to regulate its activity. However, one-half of the oocytes do form bipolar spindles, although they are short. This bipolarity suggests that some AURKA-independent aMTOC fragmentation can occur, which is not detectable in our imaging system, so that

they can form two poles or that chromatin-nucleated microtubules can cluster at their minus ends to form a pole. Another explanation could be a partial compensation by PLK4 that has been reported to be present at aMTOCs in mouse oocytes and works with AURKA to initiate microtubule nucleation [27]. We also observed short MI spindles in oocytes that lacked *Aurkb/c* where AURKA left the spindle poles and localized to chromosomes to function in the CPC [31].

In mitotic cells, phosphorylation of TPX2 by AURKA is required for MT flux, a function that maintains spindle length [62]. Therefore, in the oocytes with short, bipolar spindles, it is possible that loss of AURKA-TPX2-dependent MT flux has occurred. Moreover, the oocytes with short spindles fail to exit MI even though the APC/C is activated. Finally, another known substrate of AURKA in *Xenopus*, and likely mouse oocytes, is cytoplasmic polyadenylation element binding protein I (CPEB1) [63, 64]. When phosphorylated by AURKA, CPEB1 activates translation of maternal RNAs. In mouse, this burst of translation occurs during oocyte meiotic maturation and is required for completion of MI. Examination of this role in translation in *Aurka* KO oocytes will help explain this cell-cycle arrest.

In summary, we demonstrate that of the 3 Aurora kinases, AURKA is the only essential isoform. This is likely because AURKA regulator partner binding and substrate specificity appear to be more flexible than the other 2 kinases. Because AURKC also localizes to aMTOCs, a failure to rescue *Aurka* KO oocytes, even when overexpressed, implies that AURKC is not required for aMTOC fragmentation or for regulating TACC3, and carries out unknown aMTOC and spindle building functions. Identification of MTOC binding partners and substrates will be essential to understanding how AURKA and AURKC coordinate meiotic spindle building.

## Materials and methods

### Ethics statement

Animals were maintained following the Rutgers University Institutional Animal Use and Care Committee (Protocol 201702497), National Institutes of Health guidelines, and the policies of the Expert Committee for the Approval of Projects of Experiments on Animals of the Academy of Sciences of the Czech Republic (Protocol 43/2015). These regulatory bodies approved all experimental procedures involving the animals.

### Generation of mouse strains and genotyping

Mice possessing *loxP* sites flanking exon 3 of the *Aurka* gene [33] were obtained from (C57BL/6N; Aurkatm1c(EUCOMM)Hmgu/J) the International Mouse Phenotyping Consortium (IMPC; www.mousephenotype.org). To generate *Aurka*$^{fl/fl}$ Gdf9-Cre mice, female mice carrying the *Aurka* floxed alleles were crossed with Gdf9-Cre males (Jackson Laboratories Tg (Gdf9-icre)5092Coo/J, #011062). Mice were housed in 12–12 h light-dark cycle, with constant temperature and with food and water provided *ad libitum*. All animal experiments performed in this study were approved by the Rutgers IACUC. Genotyping for LoxP and Cre were carried out using PCR amplification. Primers for *Aurka LoxP* (Forward: 5'—CTGGATCACAGG TGTGGAGT- 3', Reverse: 5'–GGCTACATGCAGGCAAAC A—3'), and *Gdf9-Cre* (Forward: 5'—TCTGATGAAGTCAGGAAGAAC C- 3', Reverse: 5'—GAGATGTCCTTCACTCTGATT C-3', Internal control Forward: 5'—CTAGGCCACAGAATTGAAAGATCT- 3', Internal control Reverse: 5'—GTAGGTGGA AATTCTAGCATCATC C- 3') were used at 20 pMol using FastMix French PCR beads (Bulldog Bio, #25401) following manufacturer's protocol.

## Fertility trials

Sexually mature wild-type $Aurka^{fl/fl}$ and $Aurka^{fl/fl}$;Gdf9-Cre ($Aurka$ KO) female mice ages 5 to 13 weeks were continuously mated to wild type B6D2 (Jackson Laboratories B6D2F1/J, #100006) male mice with proven fertility until a total of 5 litters were produced by WT female mice. Average age of female mice at the end of the fertility trials was 6 months.

## Oocyte collection, culture, and microinjection

Fully grown, prophase I-arrested oocytes were collected from the ovaries of mice ranging in age from 3 to 12 weeks. To prevent spontaneous meiotic resumption during collection, 2.5 μM milrinone (Sigma-Aldrich #M4659) was added to minimal essential medium (MEM). To induce meiotic resumption, oocytes were cultured in milrinone-free Chatot, Ziomek, and Bavister (CZB) [65] medium in an atmosphere of 5% $CO_2$ in air at 37°C. Oocytes were matured for 7.5 hours for metaphase I experiments and 16 hours for Metaphase II experiments.

To obtain oocytes for live light-sheet time lapse imaging, prophase I-arrested oocytes were obtained as described above and oocytes were collected and microinjected in M2 medium (Sigma-Aldrich) and cultured in MEM medium (Sigma-Aldrich) supplemented with 1.14 mM sodium pyruvate (Sigma-Aldrich), 4 mg/ml bovine serum albumin (Sigma-Aldrich), 75 U/ml penicillin (Sigma-Aldrich) and 60 μg/ml streptomycin (Sigma-Aldrich), at 37°C in a 5% CO2 atmosphere. Oocytes were stained with 100 nM SiR-tubulin (Spirochrome) for microtubule visualization; SiR-tubulin was added to the culture medium. For the securin degradation analysis, a final concentration of 1 μM reversine (Sigma-Aldrich) was added to the oocytes.

For induced ovulation and collection of metaphase II eggs, female mice (>6 wks age) were injected with 5 I.U. of pregnant mare's serum gonadotropin (PMSG) (Lee Biosolutions #493–10) followed by 5 I.U. of human chorionic gonadotropin (hCG) (Sigma-Aldrich #CG5) 47 h later. 14–16 h post hCG injection, eggs were collected from the oviducts in MEM/polyvinyl-pyrrolidone media containing 3 mg/ml hyaluronidase (Sigma-Aldrich, #H3506) in MEM for 5 min. Eggs were then washed free of hyaluronidase and allowed to recover in MEM/polyvinyl-pyrrolidone media prior to fixation.

To inhibit AURKA, MLN8237 (Alisertib, Selleckchem #S1133) was added to CZB culture media at a final concentration of 1 μM. To inhibit the SAC, reversine (Cayman Chemical Research #10004412) was added to CZB culture media at a final concentration of 1 μM. Dimethyl sulfoxide (Sigma Aldrich #472301) was used as a control in the same dilution factor (1:1,000).

After removing the cumulus cells, oocytes were microinjected in M2 medium with ~10 pl of 50 ng/μl $H2b$-$mCherry$, 125 ng/μl $Egfp$-$Cdk5rap2$, 100 ng/μl $Aurka$-$Gfp$, 100 ng/μl $Aurkb$-$Gfp$, 100 ng/μl $Aurkc$-$Yfp$, 75 ng/μl $securin$-$Gfp$ cRNAs. Microinjected oocytes were cultured for 3 h in MEM medium supplemented with Milrinone to allow protein expression prior to experimental procedures.

## Plasmids

To generate cRNAs, plasmids were linearized and *in vitro* transcribed using a mMessage mMachine T3 (Ambion #AM1348) and T7 kits (Ambion #AM1344), according to manufacturer's protocol. The synthesized cRNAs were then purified using an RNAeasy kit (Qiagen #74104) and stored at -80°C. The pYX-EGFP plasmid was created by transferring T3-T7 cassette from pRNA-EGFP vector [66] into the pXY-Asc vector (NIH, Bethesda, MD, USA) using PCR cloning. The pYX-EYFP plasmid was created from pYX-EYFP plasmid by replacing coding sequence for EGFP by EYFP. AURKC coding sequence [31] was cloned by PCR into

pYX-EYFP to create pYX-AURKC-EYFP plasmid. pIVT-AURKB/C-EGFP and pGEMHE-mEGFP-mCDK5RAP2 plasmids were described previously [22, 31].

## Western blotting

A total of 100 prophase-I arrested oocytes were pooled and mixed with Laemmli sample buffer (Bio-Rad, cat #161–0737) and denatured at 95˚C for 10 min. Proteins were separated by electrophoresis in 10% SDS polyacrylamide precast gels (Bio-Rad, #456–1036). The separated polypeptides were transferred to nitrocellulose membranes (Bio-Rad, #170–4156) using a Trans-Blot Turbo Transfer System (Bio-Rad) and then blocked with 2% ECL blocking (Amersham, #RPN418) solution in TBS-T (Tris-buffered saline with 0.1% Tween 20) for at least 1h. The membranes were incubated overnight using the antibody dilution anti-AURKA (1:500; Bethyl #A300-072A), or 1 h with anti-MSY2 (1:20,000; gift from R. Schultz) as a loading control. After washing with TBS-T five times, the membranes were incubated with anti-rabbit secondary antibody (1:1000; Kindle Bioscience, #R1006) for 1 h followed with washing with TBS-T five times. The signals were detected using the ECL Select western blotting detection reagents (Kindle Bioscience, #R1002) following the manufacturers protocol. Membranes were stripped prior to loading control detection using Blot Stripping Buffer (ThermoFisher Scientific #46430) for 30 minutes at room temperature.

## Immunocytochemistry

Following meiotic maturation, oocytes were fixed in PBS containing paraformaldehyde (PFA) at room temperature (CREST, α-tubulin: 2% PFA for 20 mins; TACC3, CEP192, KIF11: 2% PFA for 30 min; C-NAP1, PLK1, phosphorylated PLK1-T210: 2% PFA + 0.1% Triton-X for 20 mins; Pericentrin, phosphorylated CDC25B-S353 and γ-tubulin, CEP215: 3.7% PFA for 1 h), PHEM (PIPES 60mM, HEPES 25mM, EGTA 10mM, and MgCl$_2$ 2mM) containing paraformaldehyde (MAD2: 2% PFA for 20 mins) or 100% Methanol for 10 min for AURKA followed by 3 consecutive washes through blocking buffer (PBS + 0.3% (wt/vol) BSA + 0.1% (vol/vol) Tween-20). Prior to immunostaining, oocytes were permeabilized for 20 min in PBS containing 0.1% (vol/vol) Triton X-100 and 0.3% (wt/vol) BSA followed by 10 min in blocking buffer. Immunostaining was performed by incubating cells in primary antibody for 1 h a dark, humidified chamber at room temperature or overnight at 4˚C followed by 3 consecutive 10 min incubations in blocking buffer. After washing, secondary antibodies were diluted 1:200 in blocking solution and the sample was incubated for 1 h at room temperature. After washing, the cells were mounted in 5 μL VectaShield (Vector Laboratories, #H-1000) with 4′, 6- Diamidino-2-Phenylindole, Dihydrochloride (DAPI; Life Technologies #D1306; 1:170).

## Antibodies

The following primary antibodies were used for immunofluorescence (IF) experiments: mouse anti α-tubulin Alexa-fluor 488 conjugated (1:100; Life Technologies #322588) AURKA (1:500; Bethyl #A300-072A), ACA (1:30; Antibodies Incorporated #15–234), phosphorylated CDC25B (1:100; Signalway Antibodies #11949), γ-tubulin (1:100; Sigma-Aldrich #T6557), MAD2 (1:100; Biolegend #PRB452C), MSY2 (1:20,000; gift from R. Schultz) [67]. TACC3 (1:100; Novus Biologicals # NBP2-67671), Phosphorylated PLK1 (1:100, BD Pharmigen #558400); Pericentrin (1:100, BD Biosciences, #611814); PLK1 (1:100, Abcam, #Ab17057); C-NAP1 (1:100, Proteintech, #14498-1-AP); CEP215 (EMD Millipore #06–1398); CEP192 (Proteintech, #18832-1-AP), KIF11 (1:50, NOVUS, # nb-500-181-ss). The following secondary antibodies were used at 1:200 for IF experiments: Anti-human-Alexa-633 (Life Technologies #A21091),

anti-mouse-Alexa-488 (Life Technologies #A11029), anti-rabbit-Alexa-568 (Life Technologies #A10042).

## Microscopy

Images were captured using a Leica SP8 confocal microscope equipped with a 40X, 1.30 N.A. oil immersion objective. For each image, optical z-slices were obtained using a 1.0 μm step with a zoom setting of 4. For comparison of pixel intensities, the laser power was kept constant for each oocyte in an experiment.

To monitor the extrusion of polar bodies, prophase I-arrested oocytes were matured *in vitro* using an EVOS FL Auto Imaging System (Life Technologies) with a 10X objective. The microscope stage was heated to 37°C and 5% $CO_2$ was maintained using the EVOS Onstage Incubator. Images were acquired every 20 min and processed using NIH Image J software.

For super-resolution microscopy we used two different microscopes: a Leica SP8 confocal microscope with Lightning module equipped with a 63X objective, 1.40 NA oil immersion objective. For each image, optical z-slices were obtained using a 0.3 μm step with a zoom setting of 4.5. A Leica SP8 Tau-STED equipped with a 93X objective, 1.3 NA glycerol immersion objective was used to image spindle poles with super resolution. The system was aligned to control any temporal and temperature dependent shift. For each image, optical z-slices were obtained using a 0.17 μm step with a zoom setting of 4.5. Excitation and depletion lasers were kept constant during image acquisition form different genotypes.

Fluorescence time-lapse image acquisitions were performed using Viventis LS1 Live light sheet microscope system (Viventis Miscoscopy Sarl, Switzerland) with a Nikon 25X NA 1.1 detection objective with 1.5 x zoom. Thirty-one 2-μm optical sections were taken with a 750 x 750-pixel image resolution using 10 min time intervals. EGFP, EYFP, mCHERRY and SiR fluorescence were excited by 488, 515, 561 and 638 nm laser lines. EGFP and EYFP emissions were detected using 525/50 (BP) and 539/30 (BP) filters, respectively. For detection of mCHERRY and SiR fluorescence, 488/561/640 (TBP) filter was used.

## Histology

Ovaries of the female mice that were in the fertility trials were fixed in Modified Davidsons fixative solution (Electron Microscopy Sciences, #6413–50) for 6–12 h and were processed by the Office of Translational Science at Rutgers University for histology services. Five μm sections of paraffin embedded ovaries were stained with Harris H/E. Ovarian images were acquired at the 1st, 5th, and 10th sections in each ovary, under a bright field microscope EVOS FL Auto Imaging System (Life Technologies) with a 20X objective and images were stitched together to project the entire ovary. Ovarian follicles were quantified using morphological criteria [68].

## Image analysis of fixed oocytes

Image J software was used to process most of the images (NIH, Bethesda, USA). For analysis, z-slices for each image were merged into a projection. Bipolar spindle length was measured between the two furthest points on both spindles using the line tool in Image J. Spindle volume was determined using the 3D reconstruction tool in Imaris software (BitPlane) freehand tool to mark precisely around the spindle. For pixel intensity analyses the average pixel intensity was recorded using the measurement tool. To define the region of the chromosomes for intensity measurements, the DNA channel (DAPI) was used as a mask. aMTOC markers, including AURKA and γ-tubulin were used to define spindle poles, and CREST was used as a kinetochore marker for pixel intensity measurements. Imaris software was used for colocalization analysis of CEP215 and Pericentrin (BitPlane). Briefly, we determined a region of interest

around the spindle pole, we set threshold for each channel and using the colocalization module we determined: Pearson coefficient which measures the covariance in the signal levels of two images; and Manders coefficients which are indicators of the proportion of the signal of one channel with the signal in the other channel over its total intensity [69, 70].

### Image analysis of live oocytes imaged by light-sheet microscopy

All image analysis was done using Fiji software [71]. For analysis of securin-EGFP degradation of the mean intensity of securin-EGFP was measured on a non-signal adjusted middle optical stack in every time frame. In every oocyte, measured mean values from each time point were normalized to the time frame with a maximum mean intensity. Calculation of destruction rate was described previously [60]. Briefly, destruction rate of securin-EGFP (h-1) was defined as the negative value of the slope of the line that can be fitted to the decreasing region of securin-EGFP destruction curve. Imaris software was also used for the analysis of aMTOC area and number over time.

### Statistical analysis

Unless stated in the legend, t-test and one-way analysis of variance (Anova) were used to evaluate the significant difference among data sets using Prism software (GraphPad Software). The details for each experiment can be found in the Results section as well as the figure legends. "Experimental n" refers to the number of animals used to repeat each experiment. Data is shown as the mean ± the standard error of the mean (SEM). $P < 0.05$ was considered significant. All statistical analysis of data from live light-sheet microscopy was done using NCSS 11 software (NCSS, LLC; Utah, USA). The type of test used are indicated in the figure legend.

### Supporting information

**S1 Fig. AURKA deletion.** Uncropped western blot detecting AURKA from prophase-I arrested wild-type (WT) and *Aurka* knockout (KO) oocytes (100 oocytes/lane). Bands at ~43kDa were included in the quantifications for AURKA signal. n = 4 animals/genotype/ experiment. Red box: Area showed in Fig 1A.
(TIF)

**S2 Fig. AURKC aMTOC localization.** Live light-sheet imaging of KO oocytes expressing histone H2B-mCHERRY (magenta), AURKC-EYFP (gray) and stained with SiR-tubulin (green). The arrows point to AURKC localization. Maximum intensity z-projections at Metaphase I. Scale bars: 10 μm.
(TIF)

**S3 Fig. Inhibition of AURKA causes spindle defects. (A)** Representative confocal images of oocytes at Metaphase I matured with MLN8237 (MLN) and immunostained with antibodies against α-Tubulin (green) and DAPI (gray). **(B)** Quantification of the percentage (%) of oocytes with different spindle phenotypes (Unpaired Students t-Test, two-tailed, * p = 0.014). **(C)** Quantification of the bipolar spindle area (Unpaired Students t-Test, two-tailed, **** p<0.0001; number of oocytes, WT: 31; KO: 23). **(D)** Quantification of the bipolar spindle length (Unpaired Students t-Test, two-tailed, **** p<0.0001; number of oocytes, WT: 30; KO: 22). Graphs show the mean ± SEM from at least 3 independent experiments.
(TIF)

**S4 Fig. *Aurka* KO oocytes have normal number of aMTOCs at prophase I.** Representative confocal images of WT and *Aurka* KO prophase I-arrested oocytes immunostained with γ-

Tubulin (magenta), α-Tubulin (green), DAPI (blue). Scale bar: 20μm.
(TIF)

**S5 Fig. PLK1 localization in *Aurka* KO oocytes. (A-B)** Representative confocal images of
oocytes from WT and KO females after nuclear envelope breakdown (NEBD) immunostained
with antibodies against PLK1 (gray), CEP192 (red), anti-centromeric antigen (ACA; cyan) and
DAPI (blue). **(C)** Quantification of PLK1 intensity at aMTOCs (Unpaired Students t-Test,
two-tailed, p = 0.389279; number of oocytes, WT: 11; KO: 9). **(D)** Quantification of PLK1
intensity at kinetochores (Unpaired Students t-Test, two-tailed, p = 0.4028; number of oocytes,
WT: 14; KO: 18).
(TIF)

**S6 Fig. Comparison of securin destruction in *Aurka* KO oocytes treated with reversine. (A)**
Live light-sheet imaging of KO oocytes expressing securin-EGFP (grey), H2B-mCherry
(magenta, chromosomes) and stained with SiR-tubulin (green, microtubules) treated with
1μM reversine. Maximum intensity z-projection images of KO oocyte arrested at MI (KO MI),
KO oocyte entering Anaphase I and extruding of polar body (KO MII), and KO oocyte enter-
ing Anaphase I but had a polar body emission error (KO PB error). Time relative to NEBD.
Scale bar = 10 μm. **(B)** Normalized intensities of cytoplasmic securin-EGFP signals. WT, KO
and KO + Reversine MI groups are same as in Fig 6D. KO + Reversine and KO + Reversine PB
error are split from KO + Reversine group in Fig 6D.
(TIF)

**S1 Movie. Movie corresponding to oocytes presented Fig 3E.**
(MOV)

**S2 Movie. Movie corresponding to oocytes presented Fig 5A.**
(MOV)

**S3 Movie. Movie corresponding to oocytes presented Fig 7C.**
(MOV)

**S4 Movie. Movie corresponding to oocytes presented S6A Fig.**
(MOV)

## Acknowledgments

The authors dedicate this manuscript to the memory of Associate Professor Dr. Petr Solc who
passed away during the revision. The authors thank Drs. Matthew McKay and Stephen A.
Murray from The Jackson Laboratory for obtaining and breeding the *Aurka* mice, Dr. Philip
Jordan at JHU for assisting with acquisition of the conditional KO mice, Ms. Marianne Polun-
nas at RU for processing the ovarian histology and Dr. Jessica Shivas at Leica Microsystems for
STED imaging acquisition. They acknowledge Richard Schultz for contributing the MSY2
antibody and members of the Schindler and Solc labs for helpful discussions.

## Author Contributions

**Conceptualization:** Petr Solc, Karen Schindler.

**Data curation:** Petr Solc.

**Formal analysis:** Cecilia S. Blengini, Patricia Ibrahimian.

**Funding acquisition:** Petr Solc, Karen Schindler.

**Investigation:** Cecilia S. Blengini, Patricia Ibrahimian, Michaela Vaskovicova, David
Drutovic.

**Methodology:** Cecilia S. Blengini, David Drutovic.

**Project administration:** Karen Schindler.

**Supervision:** Karen Schindler.

**Writing – original draft:** Cecilia S. Blengini, Patricia Ibrahimian, Petr Solc, Karen Schindler.

**Writing – review & editing:** Cecilia S. Blengini, David Drutovic, Petr Solc, Karen Schindler.

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
