## [Decision Letter · Decision Letter 0]

9 Feb 2021

Dear Karen,

Thank you very much for submitting your Research Article entitled 'Aurora kinase A is essential for meiosis in mouse oocytes' to PLOS Genetics.

The manuscript was fully evaluated at the editorial level and by independent peer reviewers. The reviewers appreciated the attention to an important topic but identified some concerns that we ask you address in a revised manuscript

We therefore ask you to modify the manuscript according to the review recommendations. Your revisions should address the specific points made by each reviewer.

[LINK]

Yours sincerely,

Paula E. Cohen

Associate Editor

PLOS Genetics

Gregory Barsh

Editor-in-Chief

PLOS Genetics

Reviewer's Responses to Questions

**Comments to the Authors:**

Reviewer #1: Review of PGENETICS-D-20-01877

Within this manuscript Cecilia S. Blengini et al. assess the role of Aurora A kinase during meiotic resumption in oocytes using a conditional knockout (CKO) mouse approach that specifically creates a knockout mutation of Aurka using the Cre recombinase transgene that is driven by the Gdf9 promoter. Within the manuscript they show that Aurka CKO mice are infertile. This infertility is not due to premature ovarian failure, but is due to the inability to segregate chromosomes during meiosis I. The authors characterize this MI failure to be due to aberrant bipolar spindle formation. The authors specifically assess PLK1 activation, microtubule organizing center (MTOC) fragmentation and loading of the major component of the liquid-like spindle domain, all of which are defective in Aurka CKO oocytes. Although these results are interesting many of the observations are not novel as small molecule inhibition and depletion approaches have previously reported these defects. Furthermore, the assessment of the MTOCs should be more thorough. An important finding from this work is that the two other Aurora family members, AURKB and AURKC cannot compensate for the loss of AURKA, which is opposite to what has been reported for AURKA, which can compensate when AURKB and AURKC are absent from an oocyte. The assessment of cohesin “cleavage” is somewhat distracting and not done using convincing assays.

Major comments:

The gamma tubulin images for WT are very hard to see in the downloaded .tiff images in Fig. 4A. (first panel). Perhaps some arrows of where real gamma tubulin signal is present or a better image would be beneficial.

The authors describe PLK1 as not being phosphorylated (pT210). However, detailed assessment of PLK1 localization and PLK1 phosphorylation is required for this study. One of the co-authors of the papers showed that phosphorylated PLK1 is present at kinetochores as well as at the MTOC (https://journals.plos.org/plosone/article?id=10.1371/journal.pone.0116783). Do the authors see defective PLK1 phosphorylation at the kinetochores as well as at the MTOC? What is the general status of PLK1 localization in the Aurka cKO? AURKC is included in the model in Fig. 7. Can AURKC phosphorylate PLK1? It would be very informative to compare a Plk1, Aurka double KO Vs the Aurka (and Plk1) CKO (or inhibition of PLK1) to determine the overlap and unique roles of these two cell cycle kinases.

In PLK1 inhibition studies by one of the co-authors gamma tubulin is absent from MTOCs. This is different from what is reported in this study. If this is the case, then some discussion is needed. In general, the contrast between Plk1 and Aurka phenotypes is needed. The authors suggest they are similar, but there are clear differences.

The authors need to assess localization of the inter-MTOC linker, C-NAP1, to determine whether Aurora A is required for timely disassociation from MTOCs.

The authors could analyze the nature of the spindles that are present as either monopolar or bipolar in more detail. Can they recover from nocodazole (or similar) treatment? Are the spindles able to reform? Where are the spindles developing from?

In Fig. 5K they show an example where PCNT is completely absent in an Aurka CKO oocyte. Is this common? Has this been quantified?

The authors should discuss the reported link between Plk4 and Aurora A that are said to cooperate in the initiation of acentriolar spindle assembly in mammalian oocytes.

The authors state that “To our surprise, although reversine-treatment 340 restored securin-EGFP destruction in Aurka KO oocytes (Fig. 6D-E), 64% of these oocytes did not enter Anaphase I and did not extrude a polar body”. Is this really a surprise given the spindle defects described earlier?

The authors state that AURKA regulates cohesin removal (although they use the word “cleavage”, which is not likely the case and was never assessed – this is a terminology that cannot be used unless assessed) in an APC/C independent manner. This is not surprising based on the role of AURKA in mitotic cells. The method of assessment of cohesin “cleavage” needs to be bolstered with further assessments beyond pixel intensity, as the role of AURKA in cohesin removal is partial. Assessment of chromatin bound cohesin Vs. free cohesin would help convince readers of this observation. Nevertheless, as it stands, this observation is somewhat distracting from the main story of the paper.

Additional comments:

The authors description of Gdf9-Cre expression and deletion of critical exons of Aurka should be better explained (lines 140 to 144).

Line 260 – “the dominate” needs to be corrected.

Error bars are missing from Fig. 3F

The authors should comment on the general morphology of the bivalents. They essentially appear to be WT for chiasma, size, etc. but this is not reported in the paper. Was there an indication of loss of association between homologs in the mutant?

At times it is hard to distinguish between channels. For example, Fig. 5A is hard to distinguish between the CDK5RAP2 signal. Same for the zoom in Fig. 5B. It would be great to have separated channels more often.

Reviewer #2: The authors made an Aurka Knockout to study the role of Aurora kinase A in oocyte meiosis and its compensatory functions with other aurora kinases. They show that the KO mutants are sterile, yet their ovaries have similar numbers of follicles. When collected from the oviduct, they found that the KO oocytes did not progress past Met I and the spindles were aberrant. Further characterization of the phenotypes showed defects in MTOC fragmentation, reduced phosphorylation of PLK1, decrease in bipolar spindle formation, reduced TACC3, and increased MAD2 localization. To study the compensatory functions of the other Aurora kinases, they overexpressed AURKB or AURKC in the AurkA KO and found that neither kinase could fully rescue the phenotypes, there were only minor rescue of the timing of bipolar spindle formation with AURKC overexpression. Interestingly, they found that abrogation of the SAC did not prevent the KO oocytes from arresting at Met I. And, further investigation showed that securin could be degraded with SAC abrogation but Rec8 was not fully cleaved. The results suggest that AURKA has a role in chromosome segregation independent of APC/C activation.

Overall, the findings in this manuscript are interesting and important for the field. The live imaging was beautiful and really enhanced the study of the phenotypes in spindle assembly to show the temporal resolution of the defects in the KO. Although some results were known from previous studies with drug inhibition or depletion of AURKA, the characterization of the KO is important to show the full extent of the defects, as some remaining AURKA activity could be present in the inhibition/depletion studies. Furthermore, there were also important results that have not been previously shown, such as the failure of AURKB and AURKC overexpression to compensate for loss of AURKA and the role of AURKA in chromosome segregation independent of APC/C activation. These results provide an important basis for future studies on the mechanism of AURKA in anaphase I onset.

Here are some suggestions for improving the manuscript:

1) In Figure 2, the graphs show a decrease in number of corpus luteum at both 2 months and 6 months in the KO compared to the WT. However, only the data at 2 months was statistically significant. The text states that there is a difference in CL and does not specify at which age. The authors should state that differences occur at 2 months and 6 months, but only the data from 2 months was significantly different from WT.

2) Figure S1 does not show the SiR-tubulin, as the figure legend states. It is hard to demonstrate that the AURKC is at the spindle poles without the Sir-tubulin.

3) When describing Fig. 4B, the authors mention that the chromosomes in the KO oocytes do not resolve from one another in early pro-Met I. However, they are basing pro-Met 1 on the timing of the wildtype oocytes. From the images and graph, it may be more accurate to say that the KO oocytes are delayed in the time of chromosome resolution because it seems that resolution occurs at later time points.

4) In the KO, do more oocytes form bipolar spindles at later time points beyond the 7hr timepoint? Are the cells with monopolar spindles delayed in bipolar spindle assembly or are they unable to form bipolar spindles?

5) When describing securin degradation, the authors fail to mention that it is substantially delayed in the KO oocytes with reversine. Although the authors are correct in highlighting the fact that securin is degraded, showing that the APC/C is activated, the delay in activation could show an important aspect to APC/C regulation by AURKA.

6) Similarly, the authors should also mention the delayed time to anaphase onset in the KO with reversine in 6G (they only mention the WT).

7) The reduction of Rec8 by 35% suggests that separase is active for Rec8 cleavage. And, if there is some separase activity, the chromosomes should eventually segregate, perhaps after a long delay. However, is the Rec8 protected from full cleavage in the KO? For example, is SGO2 protection spread to the whole chromosome (and not just the pericentromere) in the KO oocytes, preventing Rec8 cleavage?

8) Line 423 may be too strong, “….we show that PLK1 is not activated in Aurka KO oocytes”. The authors show reduced phosphorylation, not absent.

Minor typos:

Line 118 word missing

Line 167 word missing

Reviewer #3: This study by Schindler and colleagues has investigated AURKA in mouse oocytes, more specifically its unique meiotic functions that cannot be substituted for by the other family members AURKB and AURKC. The authors used mouse transgenics to specifically knockout AURKB from oocytes after completion of early meiotic events but before resumption of meiosis I. This allowed them to specifically study its role in the different stages of mouse oocyte maturation. They found that removal of AURKA causes spindle assembly defects and ultimately metaphase I arrest, effectively preventing the formation of fertilizable eggs. These phenotypes cannot be rescued by the overexpression of the other Aurora kinases while exogenous AURKA overexpression largely rescues meiotic events. The major phenotypes of spindle assembly occur at the level of microtubule nucleation by aMTOCs and failure to fragment aMTOCs and later assemble a bipolar spindle that can accurately segregate the chromosomes. Their data demonstrate AURKA activates PLK1 on aMTOCs which is required for aMTOC function in oocytes. Overall, I find this is a strong body of work that brings new knowledge to our understanding of female meiosis in mammals and I think it will be of broad interest to the readership of PLOS Genetics. However, there are a number of issues I have detailed below that the authors need to address experimentally and otherwise before this work can be published.

Major comments

- Line starting 171 – if Aurka KO ovaries had 50% reduction in the number of 172 CL in comparison to WT (Fig. 2C-F), how do the authors know the reduced fertility of AURKA KO mice is due to diminished ovulation and not because of any of the meiotic phenotypes they observe? Granted that these meiotic phenotypes can indeed cause aneuploidy and failure in embryonic development, it might be that in vivo and in a physiological environment, the ovulated oocytes can perform better than on the microscope. It may not be possible to quantify the relative contribution of meiotic defects versus failed ovulation to infertility, but the authors need to be more open about this and leave room for the alternative explanation of reduced fertility. Their finding of the meiotic defects in the absence of AURKA would still be relevant in any case. In this context, strong statements such as the one in line 177 need to be revised from being used as a premise for performing further investigations.

- F-g 3A-B – could the polar body extrusion phenotype be associated with hormonally-forced ovulation of oocytes in KO mice that would not have been ovulated? There is 50% reduction of ovulation in these animals after all.

- Lightsheet imaging of oocyte meiosis is a big effort that should provide very high temporal resolution. It is thus disappointing that the authors do not provide better movie stills in Figure 3 of the entire meiosis I process but instead opt for showing only distinct stages of meiosis I.

- The multiple stages of meiosis I in Figure 1E should be thoroughly quantified and compared between the different conditions and the method of analysis should be provided.

- What are the technical limitations mentioned in the manuscript that prevent imaging of oocyte after induction of meiosis I? This is not typical for a cell biologist studying oocyte meiosis in mice and I am concerned that MTOC convergence and stretching events preceding fragmentation are missed due to this. It would almost seem that the authors are restraining themselves from directly addressing the question by using a more advanced imaging technology when they could have captured the whole process of MTOC coalescence, stretching and fragmentation using an ‘inferior’ confocal system that was used to describe this process originally (ref 10).

- Overall, from the supplementary movies it appears that the major spindle assembly defects could be MTOC lack of stretching/fragmentation followed by bipolarisation defects. Can the authors perform better quantification of stretching/fragmentation by taking advantage of the lightsheet imaging they have done (e.g. volume measurement over time of MTOCs)?

- These two processes are motorized – stretching/fragmentation by dynein and bipolarisation by kinesin-5. How are these motors affected in AURKA KO oocytes? while the PLK data demonstrate that aMTOCs are largely inactive (low microtubule mass in prometaphase), this should not fully affect at least bipolarisation since chromatin-mediated microtubule nucleation can eventually assemble microtubules. This suggests other spindle assembly factors such as kinesin-5 may also be affected in AURKA KO oocytes.

- How does loss of AURKA affect cohesion? Rec8 is obviously not cleaved as much in these oocytes but this observation is rather descriptive at this stage and leaves the reader with many mechanistic questions that are not addressed in the current study. It might be beneficial to significantly shorten this section or even leave it out as at this point it does not provide much insight about AURKA function. Furthermore, in reversine treated KO oocytes, even though securing is destroyed, is the APC/C really active? Is separase active? What is the status of Sgo2, PP2A and other proteins involved in this complex phase of anapase I? Unless the authors wish to go into mechanistic details that address these issues, the Rec8 section should be revised. I would say this is a topic for an independent study and takes attention away from the unique function of AURKA at aMTOCs.

- I would argue that the mouse oocyte spindle pole being a liquid-like or phase-separated structure is still a model that requires further research before it is accepted in the field. I do see why the authors are referring to it as such since the model was very recently proposed. However, it is anecdotal at this point and it would be better to remove this term to avoid confusion of non-expert readers. Besides, referring to the spindle pole as LISD just adds to the confusion. The paragraph starting at line 301 is sufficient to discuss this without going into too much detail and featuring phase separation in the abstract and introduction. TACC3 presence on the spindle poles is not by itself evidence that the phase separated nature of the spindle pole, if this is indeed the nature of oocyte spindles, is unaffected in AURKA oocytes. To put it simply, there is only one study proposing this model and without further experimental validation by independent studies, we should refrain from accepting it. Again, this is anecdotal to the current manuscript’s conclusion and rephrasing this should not affect the main message of the paper.

Minor comments

- Line 78 – centrioles ‘are’ lost. Even better to use the term ‘eliminated’ instead since at least in other species this is an active process. It would also be good to cite here how centriole elimination in oocytes is conserved across several species from flies, echinoderms to mammals.

- The authors should follow this up with introducing MTOCs as aMTOCs instead and use this term throughout the manuscript. This is what makes these structures special in oocytes of many species.

- Line 82 – better to stick here with more widely used terms such as MTOC coalescence here – decondense is more typically used with chromosome morphology in chromosome segregation research and that MTOCs decondense might sound strange. The process also starts with MTOCs migrating onto the nucleus from the cytoplasm where they coalesce, stretch and fragment (demonstrated in ref 10).

- Line 89 – the SAC can be fulfilled by ‘inappropriate’ attachments that still cause tension so for accuracy, better to write ‘until all kinetochores are attached to microtubules’.

- Figure 1 – please refrain from quantifying depletion from ECL blots, which are by definition not quantitative. If quantification is needed, this should be done from fluorescent westerns, but it is not. The blots in figure 1A are pretty clear that the deletion has worked. Could the authors also provide uncropped blots in the supplementary data as well?

**Have all data underlying the figures and results presented in the manuscript been provided?**

Reviewer #1: None

Reviewer #2: Yes

Reviewer #3: Yes

PLOS authors have the option to publish the peer review history of their article (what does this mean?). If published, this will include your full peer review and any attached files.

Reviewer #1: No

Reviewer #2: No

Reviewer #3: No

---

## [Editor Report · Decision Letter 1]

8 Apr 2021

Dear Karen,

We are pleased to inform you that your manuscript entitled "Aurora kinase A is essential for meiosis in mouse oocytes" has been editorially accepted for publication in PLOS Genetics. Congratulations on a beautiful piece of work!

Best wishes,

Paula E. Cohen

Associate Editor

PLOS Genetics

Gregory Barsh

Editor-in-Chief

PLOS Genetics

Comments from the reviewers (if applicable):

**Data Deposition**

http://datadryad.org/submit?journalID=pgenetics&manu=PGENETICS-D-20-01877R1

**Press Queries**

---

## [Editor Report · Acceptance letter]

20 Apr 2021

PGENETICS-D-20-01877R1 

Aurora kinase A is essential for meiosis in mouse oocytes 

Dear Dr Schindler, 

We are pleased to inform you that your manuscript entitled "Aurora kinase A is essential for meiosis in mouse oocytes" has been formally accepted for publication in PLOS Genetics! Your manuscript is now with our production department and you will be notified of the publication date in due course.

With kind regards,

Katalin Szabo

PLOS Genetics

On behalf of:
